# Differentially Private Space-Efficient Algorithms for Counting Distinct Elements in the Turnstile Model

Rachel Cummings [* 1]  Alessandro Epasto [* 2]  Jieming Mao [* 2]  Tamalika Mukherjee [* 1]  Tingting Ou [* 1]
Peilin Zhong [* 2]

## Abstract

The *turnstile* continual release model of differential privacy captures scenarios where a privacy-preserving real-time analysis is sought for a dataset evolving through additions and deletions. In typical applications of real-time data analysis, both the length of the stream $T$ and the size of the universe $|\mathcal{U}|$ from which data come can be extremely large. This motivates the study of private algorithms in the turnstile setting using space sublinear in both $T$ and $|\mathcal{U}|$. In this paper, we give the first sublinear space differentially private algorithms for the fundamental problem of counting distinct elements in the turnstile streaming model. Our algorithm achieves, on arbitrary streams, $\tilde{O}_\eta(T^{1/3})$ space and additive error, and a $(1 + \eta)$-relative approximation for all $\eta \in (0, 1)$. Our result significantly improves upon the space requirements of the state-of-the-art algorithms for this problem, which is linear, approaching the known $\Omega(T^{1/4})$ additive error lower bound for arbitrary streams. Moreover, when a bound $W$ on the number of times an item appears in the stream is known, our algorithm provides $\tilde{O}_\eta(\sqrt{W})$ additive error, using $\tilde{O}_\eta(\sqrt{W})$ space. This additive error asymptotically matches that of prior work which required instead linear space. Our results address an open question posed by (Jain et al., 2023) about designing low-memory mechanisms for this problem. We complement these results with a space lower bound for this problem, which shows that any algorithm that uses similar techniques must use space $\tilde{\Omega}(T^{1/3})$ on arbitrary streams.

[1]Department of Industrial Engineering and Operations Research, Columbia University, New York, USA [2]Google Research, New York, USA. Correspondence to: Tamalika Mukherjee <tm3391@columbia.edu>.

*Proceedings of the 42nd International Conference on Machine Learning*, Vancouver, Canada. PMLR 267, 2025. Copyright 2025 by the author(s).

## 1. Introduction

Data streaming applications in which one needs to track specific statistics over a period of time occur in many real-world scenarios including digital advertising, network monitoring, and database systems. Given the sheer volume of data collected and processed in these online settings, significant work has been dedicated to the design of efficient algorithms to track and release useful statistics about the data stream, at every timestep. A key goal in this area is the design of space-efficient algorithms, i.e., algorithms that do not require storing the entire stream in memory.

In many applications, the data being collected is not only massive, but also contains sensitive and personal user information. In this case, formal privacy protections are often required to ensure that the data released by the algorithm does not inadvertently leak protected information about individual users. Differential privacy (DP) (Dwork et al., 2006) has emerged as the de facto gold-standard in privacy-preserving data analysis to address such concerns.

We focus on the *turnstile* model of *continual release* in differential privacy. In this model, we are given a stream $x = (x_1, \ldots, x_T)$, where each $x_t$ can be an insertion $(+u)$ or deletion $(-u)$ of some data item $u$ from a fixed universe $\mathcal{U}$, or $\perp$ (no update), and the goal is to release a statistic of interest at every timestep $t \in [T]$, in a differentially private manner. An algorithm $\mathcal{A}$ is $(\varepsilon, \delta)$-*differentially private* if for any neighboring input streams $x \sim x'$ and any output set $O$,

$$\Pr[\mathcal{A}(x) \in O] \le e^\varepsilon \Pr[\mathcal{A}(x') \in O] + \delta.$$

In the continual release model, the output $\mathcal{A}(x)$ refers to the entire output history of the algorithm $\mathcal{A}$ over stream $x$ at every timestep. We consider event-level privacy, i.e., the input streams $x \sim x'$ are neighboring if they differ in at most one timestep.

In this paper, we study the fundamental problem of counting distinct elements in the turnstile model under continual release. Starting from the seminal works of (Flajolet & Martin, 1985; Alon et al., 1999), there is a rich literature of low space algorithms for the problem of counting distinct elements in the non-DP streaming world (e.g., see references within (Muthukrishnan, 2005)).

While space-efficient DP algorithms for this problem have been recently studied in the (1) one-shot model (where the algorithm produces an output once at the end of the stream) (Pagh & Stausholm, 2021; Wang et al., 2022; Dickens et al., 2022; Hehir et al., 2023; Smith et al., 2020; Stanojevic et al., 2017; Braverman et al., 2023; Blocki et al., 2023), and (2) insertion-only continual release model (where the algorithm outputs at every timestep, but no deletions are allowed) (Epasto et al., 2023; Dwork et al., 2010; Chan et al., 2011; Bolot et al., 2013; Ghazi et al., 2023), the existence of a private space-efficient algorithm in the more general turnstile model under continual release has been largely unexplored. Notably, even without any space constraints, recent work (Jain et al., 2023) has shown that designing DP algorithms for this problem in the general turnstile model is more challenging and can incur significantly more additive error, i.e., polynomial in $T$ in the turnstile model vs. polylogarithmic in $T$ in the insertion-only model. Understanding whether this fundamental problem can be solved in the turnstile model under continual release with reasonable accuracy and efficiency is an open problem. We address this gap by presenting the first sublinear space algorithm (sublinear in $T$ and $|\mathcal{U}|$) for counting distinct elements.

For counting distinct elements in an input stream $x$ under continual release, one strategy used by prior work (Epasto et al., 2023; Jain et al., 2023) is to approximately track the summation over the stream $s_x \in \{-1, 0, 1\}^T$ with the binary tree mechanism (Chan et al., 2011; Dwork et al., 2010), where $s_x(t)$ is the difference in the number of distinct elements at timestep $t - 1$ and $t$. Note that $\sum_{i \leq t} s_x(i)$ precisely gives the number of distinct elements at timestep $t$. In the insertion-only setting, it is easy to observe that the sensitivity of the summation stream $s_x$ is bounded by a constant (Epasto et al., 2023; Bolot et al., 2013). In the turnstile model, (Jain et al., 2023) observed that a change in the stream $x$ can cause $\Omega(T)$ changes to the corresponding summation stream $s_x$. In particular, the sensitivity of the summation stream $s_x$ in the turnstile model depends on the number of times an input stream item switches between being present to absent or vice versa — this property is called the *flippancy* of the item. (Jain et al., 2023) designed DP algorithms for counting distinct elements where the additive error scales with the flippancy of the stream, denoted $w_x$ (i.e., maximum flippancy over all items in the universe for a fixed stream). More specifically, the algorithm imposes a (provable) bound on the maximum flippancy of the stream.

However, flippancy is ill-suited to design low-space algorithms due to its inherently *stateful* nature (i.e., whether an element flips at time $t$ depends on all prior events of that element). For this reason, we introduce the related, but *stateless*, notion of *occurrency* (see Definition 1.1) which measures the maximum number of times an item appears in the stream. Using this occurrency measure, we design space-

efficient DP algorithms for counting distinct elements with error and space that scales with the maximum occurrency.

**Definition 1.1** (Occurrency). Given a stream $x$ and an element $u \in \mathcal{U}$, the *occurrency* of $u$ in $x$, denoted $\operatorname{occur}(u, x)$, is the number of times an element $u$ appears (either as insertion or as deletion) in the stream $x$. The occurrency of the stream $x$, denoted $W_x$, is $\max_{u \in \mathcal{U}} \operatorname{occur}(u, x)$.

We remark that for any stream $x$ of size $T$, $W_x \leq T$; however, in many instances, the occurrency is much smaller, and can be as low as $T / |\mathcal{U}|$. For example, consider the case of an online learning platform that wants to estimate the number of activities being actively performed at the moment by users. A user can begin one activity (e.g., initiating an homework assignment for a course) and then terminate it when they are done (e.g., completing the assignment).[1] Each event in this example stream represents initiations/terminations of a given activity (where $+u$ is an initiation of an activity $u$, and $-u$ is a termination). In this case, occurrency can be bounded by twice the maximum number of students in a class, which is clearly $\ll T$ (total number of events).

In this paper, we first design a DP algorithm for counting distinct elements that, when provided with a stream promised to have occurrency bounded by $W$, has an error and space that scales as a function of the promised bound $W$. When a bound on occurrency is known, as in the example above, this first algorithm can be used as is and will satisfy DP guarantees. However, in many cases an occurrency bound may not be known. For this reason, we also design a more general algorithm that for an *arbitrary* stream of unknown occurrency, *imposes* a bound on occurrency for a subset of the stream. Combining these results, we prove our main result: a low-space algorithm with *unconditional* DP guarantees for arbitrary streams — i.e., the algorithm is *truly private* in the sense of (Jain et al., 2023). Finally, we give a matching space lower bound showing that any algorithm using a technique based on bounding occurrency (or flippancy) cannot hope to achieve better space bounds.

### 1.1. Our Contributions

The main result of our paper is the design of the first sublinear space DP algorithms for the problem of counting distinct elements in the turnstile model under continual release. Before presenting our results, we first define some useful notation and terminology. We say $v$ is an $(\alpha, \beta)$-approximation of a function $f$ on input stream $x$ if $(1/\alpha)f(x) - \beta \leq v \leq \alpha f(x) + \beta$.

We first present a DP algorithm for the problem of counting distinct elements, where the input stream has a promised

---

[1]In this simple example, for clarity, we assume a user can only participate in the activity once. Our algorithms make no such assumption.

upper bound on its occurrency. In such cases, our DP algorithm achieves space and error guarantees that only have a sublinear dependency on occurrency as stated below.[2]

This algorithm utilizes a low-space dictionary data structure denoted KSET, which was introduced by (Ganguly, 2007) to estimate the number of distinct elements in the non-DP turnstile streaming setting. KSET supports the insertion and deletion of data items and returns (with high probability) the set of items $S$ present in the dictionary as long as $|S| \leq k$. If the number of elements exceeds $k$ or there is a collision where two elements are assigned the same cell in the KSET, then it "fails" by outputting NIL.

**Theorem 1.2** (Informal, Corollary 3.1). *For all $\varepsilon, \eta > 0$ and $\delta \in (0,1)$ and streams of length $T \in \mathbb{N}$, given a promised occurrency bound of $W_x$, there exists an $(\varepsilon, \delta)$-DP algorithm in the turnstile model under continual release that outputs a $(1 + \eta, \tilde{O}_{\varepsilon,\delta,\eta}(\sqrt{W_x}))$-approximation to the number of distinct elements in the stream, using space $\tilde{O}_{\varepsilon,\delta,\eta}(\sqrt{W_x})$.*

We remark that in this result, DP is guaranteed only when the input streams do not violate the promised occurrency upper bound. Our next result allows us to remove this assumption and provide a DP algorithm for streams with unbounded occurrency by creating a *blocklist* of high occurrency items to effectively ignore them in future timesteps. Before presenting our main result, we first define the problem of blocklisting high occurrency items and show a near optimal space algorithm for this problem.

Let $\mathbf{blocklist_{occ}(W)}$ be the problem of outputting 0 at timestep $t$ when the current input element has occurrency $< W$ (up to the step before), and 1 otherwise. An algorithm for solving $\mathbf{blocklist_{occ}(W)}$ reports a *false negative* if the element at timestep $t$ has occurrency $\geq W$ but the algorithm outputs 0; it reports a *false positive* if the element at timestep $t$ has occurrency $< W$ but the algorithm outputs 1.

**Theorem 1.3** (Informal, Corollary E.3 and Theorem E.2). *There exists an algorithm that, with high probability, has no false negatives and bounded false positives for $\mathbf{blocklist_{occ}(W)}$ and uses space $\tilde{O}(T/W)$. Moreover, for any $W > 0$, any algorithm that solves $\mathbf{blocklist_{occ}(W)}$ with the same false positive bound (and no false negatives) needs space $\tilde{\Omega}(T/W)$.*

This space lower bound applies to any algorithm (even exponential time ones) and implies that the space bounds achieved by our final algorithms are optimal up to log factors (among algorithms using the blocklisting approach).

Finally, our main result gives a DP algorithm for counting

distinct elements with no assumptions on the occurrency of the input stream. Our algorithm uses the blocklisting technique described above with the optimal choice of occurrency $W_x = T^{2/3}$, combined with the KSET data strucure to achieve sublinear space and non-trivial additive error.

**Theorem 1.4** (Main (Informal), Corollary 3.2). *For all $\varepsilon, \eta > 0$ and $\delta \in (0,1)$ and streams of length $T \in \mathbb{N}$, there exists an $(\varepsilon, \delta)$-DP algorithm in the turnstile model under continual release that outputs a $(1 + \eta, \tilde{O}_{\varepsilon,\delta,\eta}(T^{1/3}))$-approximation to the number of distinct elements in the stream, using space $\tilde{O}_{\varepsilon,\delta,\eta}(T^{1/3})$.*

**Comparison to prior work.** We observe that our results are the first to address the open question posed by (Jain et al., 2023) regarding the existence of accurate, private, and *low-memory* mechanism for counting distinct elements in turnstile streams. When the stream has a promised flippancy bound $w_x$ rather than a promised occurrency bound, the informal Theorem 1.2 (and the formal counterpart Corollary 3.1) can be restated equally in terms of the flippancy of the stream. More precisely, our algorithm provides a $1 + \eta$ multiplicative approximation with $\tilde{O}_{\varepsilon,\delta,\eta}(\sqrt{w_x}))$ additive error, using space $\tilde{O}_{\varepsilon,\delta,\eta}(\sqrt{w_x})$.[3] This additive error matches (neglecting lower order terms) the additive error $\tilde{O}_{\varepsilon}(\min(\sqrt{w_x}, T^{1/3}))$ in the algorithm of (Jain et al., 2023)[4] which is achieved however with $\Omega(T)$ space. (This is because for the regime of $w_x \geq T^{2/3}$ one can use the unbounded occurrency algorithm in our paper to obtain additive error $\tilde{O}_{\varepsilon,\delta,\eta}(T^{1/3})$). When flippancy is unbounded, our additive error is always $\tilde{O}_{\varepsilon,\delta,\eta}(T^{1/3})$. This is close to the lower bound of $\tilde{\Omega}_{\varepsilon}(T^{1/4})$ from (Jain et al., 2023) for private algorithms for the problem in streams of arbitrary large flippancy. Closing this gap is an interesting open problem.

## 1.2. Related Work

Our work on designing space-efficient turnstile streaming algorithms in the DP continual release setting is related to several topics in the areas of (non-private) space-efficient streaming algorithms, DP continual release, and DP streaming algorithms. Additional related work can be found in Appendix A.1.

**Space-efficient streaming algorithms.** The streaming model of computation (Flajolet & Martin, 1985) is a well-known abstraction for efficient computation on large-scale data. In this model, data are received one input at a time, and the algorithm designer seeks to design algorithms that compute a solution on-the-fly while using limited space and time. In the area of non-private computation, a vast literature

---

[2]We present our informal results in terms of $(\varepsilon, \delta)$-DP, however our formal theorems are stated in terms of $\rho$-zCDP. We can use Theorem A.2 to convert $\rho$-zCDP to $(\varepsilon, \delta)$-DP.

[3]We omit the proof for simplicity as it is follows the same steps as our proof in terms of occurrency.

[4]Sublinear space requires a multiplicative approximation factor in our algorithm.

has been developed over the past decades (Morris, 1978; Flajolet & Martin, 1985; Flajolet et al., 2012; Alon et al., 1996) for addressing a variety of problems ranging from classical streaming computations (such as heavy hitters and frequency moments) (Flajolet & Martin, 1985; Flajolet et al., 2012; Durand & Flajolet, 2003; Cormode & Muthukrishnan, 2005; Misra & Gries, 1982), to solving combinatorial optimization problems (McGregor & Vorotnikova, 2018; Huang & Peng, 2019; Charikar et al., 2003). Two key questions in the literature are whether it is possible to obtain space sublinear in the number of updates (Flajolet et al., 2012) and whether it is possible to handle increasingly more dynamic updates (i.e., insertions-only, sliding window (Datar et al., 2002), and fully-dynamic streams (McGregor & Vorotnikova, 2018)). In the (non-private) streaming literature, there is a well-developed theoretical understanding for interplay between the dynamicity of the stream, the accuracy achievable, and space bounds required, for a vast array of algorithmic problems (Woodruff, 2004).

**DP turnstile model.** All previously mentioned work does not consider the fully dynamic streaming setting, where items can also be removed from the stream. The DP turnstile continual release model is the private equivalent of the fully-dynamic streaming model, and has received substantially less attention in the literature. Counting distinct elements in the turnstile continual release model was only recently studied for the first time by (Jain et al., 2023). For a stream with flippancy bound $w$, they give an $(\varepsilon, \delta)$-DP mechanism with additive error $O(\varepsilon^{-1}\sqrt{w}\,\mathrm{poly}\log(T)\sqrt{\log(1/\delta)})$ and space $\Omega(T)$. They also show a lower bound of $\Omega(\min(w, T^{1/4}))$ on the additive error for any DP mechanism for this problem. In a concurrent work, (Henzinger et al., 2023) also studied this problem under a restricted variant of the turnstile model where items are guaranteed to be present with cardinality at most 1 at any time (i.e., multiple insertions of the same element are ignored). For this setting, they give an $(\varepsilon, 0)$-DP algorithm with additive error $\tilde{O}(\sqrt{\varepsilon^{-1}K\log(T)})$, where $K$ is the total number of insertions and deletions, and a nearly matching lower bound. Contrary to the non-private literature, theoretical understanding of space efficiency in private dynamic streaming algorithms is very limited. No prior work has designed differentially private sublinear-space algorithms for the foundational problem of counting distinct elements in the turnstile continual release model.

## 2. Preliminaries

We consider an input stream $x_1, \ldots, x_T$ of length $T$, coming from universe $\mathcal{U}$, such that each $x_i \in \mathcal{U} \cup \{\bot\}$. We assume that $|\mathcal{U}| = \mathrm{poly}(T)$, as is standard in streaming literature (Chakrabarti, 2012). This assumption allows for simplified lower bound on space, since storing a single item

from the universe requires $O(\log(|\mathcal{U}|)) = \log(T))$ bits.

First, we recall the definition of differential privacy (DP) on streams. Neighboring streams $x$ and $x'$, denoted $x \sim x'$, differ in the stream elements at most one timestep.

**Definition 2.1** (Differential Privacy (Dwork et al., 2006))**.** Given privacy parameters $\varepsilon > 0$ and $\delta \in [0, 1)$, an algorithm $\mathcal{A}$ is $(\varepsilon, \delta)$-DP if for any neighboring streams $x \sim x'$ and any output set $O$, $\Pr[\mathcal{A}(x) \in O] \leq e^\varepsilon \Pr[\mathcal{A}(x') \in O] + \delta$.

When $\delta = 0$, this is known as pure DP, when $\delta > 0$ and it is known as approximate DP. In the continual release setting, the output of $\mathcal{A}(x)$ is the entire $T$-length output over the stream $x$ at every timestep.

Our privacy analysis will primarily use zero-concentrated differential privacy (zCDP), which is a slight variant of the standard DP definition. We present the definition of zCDP, its composition properties, and its relationship to $(\varepsilon, \delta)$-DP in the one-shot setting, where the entire stream is processed and only one output is released at the end of the stream.

**Definition 2.2** (zero-Concentrated Differential Privacy (zCDP) (Bun & Steinke, 2016))**.** Given a privacy parameter $\rho > 0$, a randomized algorithm $\mathcal{A}$ satisfies $\rho$-zCDP if for all pairs of neighboring streams $x \sim x'$ and all $\alpha > 1$,

$$\mathcal{D}_\alpha(\mathcal{A}(x)\|\mathcal{A}(x')) \leq \rho\alpha,$$

where $\mathcal{D}_\alpha(P\|Q) = \frac{1}{\alpha-1} \log\left(E_{y\sim P}\left[\frac{P(y)^{\alpha-1}}{Q(y)^{\alpha-1}}\right]\right)$ is the Renyi divergence of order $\alpha$ between probability distributions $P$ and $Q$.

There is also a relaxation of zCDP, known as *approximate zCDP*, which is analogous to the relaxation between pure DP and approximate DP.

**Definition 2.3** (Approximate zCDP (Bun & Steinke, 2016))**.** Given privacy parameters $\rho > 0$ and $\delta \in (0, 1)$, a randomized algorithm $\mathcal{A}$ satisfies $\delta$-approximate $\rho$-zCDP if for all pairs of neighboring streams $x \sim x'$, there exist events $E$ (which depends on $\mathcal{A}(x)$) and $E'$ (which depends on $\mathcal{A}(x')$) such that $\Pr[E] \geq 1 - \delta$, $\Pr[E'] \geq 1 - \delta$, and for all $\alpha \in (1, \infty)$,

$$D_\alpha(\mathcal{A}(x)|_E\|\mathcal{A}(x')|_{E'}) \leq \rho\alpha \,\vee\, D_\alpha(\mathcal{A}(x')|_{E'}\|\mathcal{A}(x)|_E) \leq \rho\alpha,$$

where $\mathcal{A}(x)|_E$ is the distribution of $\mathcal{A}(x)$ conditioned on $E$.

The privacy parameters of zCDP compose, similar to the composition guarantees of DP, and it is possible to translate between the guarantees of zCDP and DP. Both of these are shown in Appendix A.2, along with other additional details on zCDP.

## 3. Estimating the Number of Distinct Elements

In Section 3.1, we first present our DP algorithm (Algorithm 1), which outputs an approximation of the number of

distinct elements in the stream in a continual release manner. We then present our main results for this algorithm in Section 3.2, namely its privacy, accuracy, and space guarantees, will details of the analysis deferred to Section 4.

### 3.1. Algorithm and Description

Our algorithm CountDistinct (Algorithm 1) for counting the number of distinct elements in a stream uses three main ingredients: (1) a KSET data structure which we use to store distinct elements from the stream $x$ with low space, (2) a binary-tree mechanism BinaryMechanism-CD for estimating the summation stream $s_x(t) \in \{-1, 0, 1\}^T$ which we obtain by comparing the cardinality of the distinct element set returned by the KSET data structure at timesteps $t - 1$ and $t$, and (3) a blocklist $\mathcal{B}$ which with high probability, stores all items whose occurrency is too large.

At a high-level, CountDistinct executes the following process on different subsamples. At each timestep $t \in [T]$, if the data element $x_t$ is non-empty, then the COUNTING-KSET subroutine updates the KSET data structure with $x_t$, and obtains the current set of distinct elements from the KSET (if the KSET does not fail; we discuss later how to deal with failures in the KSET as this case requires more care). If the item corresponding to $x_t$ is present in the blocklist $\mathcal{B}$, then the COUNTING-KSET subroutine does not update the KSET data structure with $x_t$. Next, the algorithm computes $s_x(t)$ to be the difference in counts between the current distinct elements set and the previously stored set, and feeds $s_x(t)$ into BinaryMechanism-CD, which produces a differentially privat count of the number of distinct elements $\hat{s}_x(t)$. Finally $\hat{s}_x(t)$ is compared to a fixed threshold $\tau$. If $\hat{s}_x(t)$ is greater than $\tau$, then COUNTING-KSET returns TOO-HIGH, otherwise it returns $\hat{s}_x(t)$ to the main algorithm. The algorithm then performs the block-listing step, which adds the current item to the blocklist $\mathcal{B}$ with probability $p$ — this ensures that the elements with high occurrence are not likely to be considered by our algorithm in future iterations. This process is executed in $\log(T)$ parallel instances of COUNTING-KSET with different sampling rates using a hash function, in order to ensure that at least one sampling rate yields a good approximation to the number of distinct elements.

More concretely, Algorithm 1 takes in a boolean flag $ob$ as input, indicating whether there is a promised bound on the occurrence of the stream — if $ob$ is true, the algorithm will operate under the assumption that the input stream has an occurrence upper bounded by $W$, and thus does not employ the blocklisting technique. If $ob$ is false, the algorithm imposes an internal bound of $W = T^{2/3}$ and executes the blocklisting procedure, in which it fixes a sampling probability $p$ (in Line 16), and every time an element occurs, the element is sampled with probability $p$ to be stored in a

---

**Algorithm 1** CountDistinct

**Require:** Stream $x_1, \ldots, x_T \in \mathcal{U}$, relative error $\eta \in (0, 0.5)$, privacy parameter $\rho$, failure probability $\beta$, boolean $ob$ that signals if we have an occurrency bound on elements, occurrency bound $W$

1: Let $L \leftarrow \lceil \log(T) \rceil$, $\lambda \leftarrow 2 \log(40L/\beta)$
2: **if** $ob$ is true **then**
3: $\quad \gamma = \sqrt{\frac{4(W+1)(\log T+1)^3 \log(10(\log T+1)/\beta)}{\rho}}$
4: **else**
5: $\quad \gamma = \sqrt{\frac{4(T^{2/3}+1)(\log T+1)^3 \log(10(\log T+1)/\beta)}{\rho}} + 3T^{1/3} \log(T^{1/3} \lceil \log T \rceil / \beta)$
6: **end if**
7: Let $g : \mathcal{U} \rightarrow [L]$ be a $\lambda$-wise independent hash function; for every $a \in \mathcal{U}$, $i \in [L]$, $\Pr[g(a) = i] = 2^{-i}$, $\Pr[g(a) = \perp] = 2^{-L}$
8: Initialize empty streams $\mathcal{S}_1, \ldots, \mathcal{S}_L$ {$\mathcal{S}_i$ is the stream of noisy distinct counts and TOO-HIGHs}
9: Initialize COUNTING-KSET$_1, \ldots,$ COUNTING-KSET$_L$ with occurrency bound $W$ if $ob = true$ or occurrency bound $T^{2/3}$ if $ob = false$
10: Initialize blocklist $\mathcal{B} = \emptyset$
11: **for** update $x_t$ **do**
12: $\quad$ **for** $i \in [L]$ **do**
13: $\quad\quad$ **if** $x_t \neq \perp$ and $g(x_t) = i$ **then**
14: $\quad\quad\quad \mathcal{S}_i[t] =$ COUNTING-KSET$_i$.Update$(x_t, \mathcal{B})$ {see Algorithm 2}
15: $\quad\quad\quad$ **if** $x_t \notin \mathcal{B}$ and $ob$ is false **then**
16: $\quad\quad\quad\quad$ Add $x_t$ to $\mathcal{B}$ with probability $p = \frac{\log(T^{1/3}L/\beta)}{T^{2/3}}$
17: $\quad\quad\quad$ **end if**
18: $\quad\quad$ **else**
19: $\quad\quad\quad \mathcal{S}_i[t] =$ COUNTING-KSET$_i$.Update$(\perp, \mathcal{B})$ {see Algorithm 2}
20: $\quad\quad$ **end if**
21: $\quad$ **end for**
22: $\quad$ **Output:** $\mathcal{S}_i[t] \cdot 2^i$ for the largest $i \in [L]$ such that $\mathcal{S}_i[t]$ is not TOO-HIGH and $\mathcal{S}_i[t] \geq \max\{\gamma/\eta, 32\lambda/\eta^2\}$. (If such $i$ does not exist, **output** 0.)
23: **end for**

---

blocklist $\mathcal{B}$.

Algorithm 1 uses a hash function to generate multiple parallel substreams $i \in [L]$ of the input stream (see Line 7) where $L = \lceil \log(T) \rceil$, all subsampled with different sampling rates. The different sampling rates ensure that at least one sampling rate will yield a good approximation of the number of distinct elements in the original stream.

For each instance, $i \in L$, Algorithm 1 initializes the DP subroutine COUNTING-KSET$_i$ (Algorithm 2), which uses two key subroutines:

**Algorithm 2** COUNTING-KSET

---

**Require:** Stream update $x_1, \ldots, x_T \in \mathcal{U}$, relative error $\eta \in (0, 0.5)$, privacy parameter $\rho$, failure probability $\beta$, substream index $i$, occurrence bound $W$ on elements, blocklist $\mathcal{B}$

1: Initialize $\tau = 16 \max\{\gamma/\eta, 32\lambda/\eta^2\} + 2\sqrt{2}\frac{(\log T+1)^{3/2}\sqrt{W \log(20T\lceil \log T\rceil/\beta)}}{\sqrt{\rho}}$

2: Initialize $k = 16 \max\{\gamma/\eta, 32\lambda/\eta^2\} + 4\sqrt{2}\frac{(\log T+1)^{3/2}\sqrt{W \log(20T\lceil \log T\rceil/\beta)}}{\sqrt{\rho}}$

3: Initialize BinaryMechanism-CD$_i$ with parameters: privacy parameter $\rho/L$ and occurrence bound $W$

4: Initialize KSET$_i$ data structure with parameters: capacity $k$ and failure probability $\beta/(2TL)$

5: Initialize $F_{i,\mathsf{last}} = 0$, $t_{last} = 0$

6: **Update**$(x_t, \mathcal{B})$:

7: **for** update $x_t$ **do**

8:     **if** $x_t \neq \perp$ and $x_t \notin \mathcal{B}$ **then**

9:         KSET$_i$.Update$(x_t)$

10:     **end if**

11:     Let $S_i \leftarrow$ KSET$_i$.ReturnSet {Only keep the elements but not their counts }

12:     **if** $S_i \neq$ NIL **then**

13:         $t_{\mathsf{diff}} = t - t_{\mathsf{last}}$

14:         $\mathsf{diff} = |S_i| - F_{i,\mathsf{last}}$

15:         **for** $j = 1$ to $|\mathsf{diff}|$ **do**

16:             **if** $\mathsf{diff} > 0$ **then**

17:                 $\hat{s}_i \leftarrow$ BinaryMechanism-CD$_i$.Update$(1)$

18:             **else if** $\mathsf{diff} < 0$ **then**

19:                 $\hat{s}_i \leftarrow$ BinaryMechanism-CD$_i$.Update$(-1)$

20:             **end if**

21:         **end for**

22:         **for** $j = 1$ to $t_{\mathsf{diff}} - |\mathsf{diff}|$ **do**

23:             $\hat{s}_i \leftarrow$ BinaryMechanism-CD$_i$.Update$(0)$

24:         **end for**

25:         Save $F_{i,\mathsf{last}} \leftarrow |S_i|$

26:         $t_{\mathsf{last}} = t$

27:     **end if**

28:     **if** $\hat{s}_i > \tau$ or $S_i =$ NIL **then**

29:         Return TOO-HIGH

30:     **else**

31:         Return $\hat{s}_i$

32:     **end if**

33: **end for**

---

1. KSET (Algorithm 4): The $k$-set structure is a dictionary data structure that supports insertion and deletion of data items and either returns, with high probability, the set of items $S$ that are present in the dictionary if $|S| \leq k$, or returns NIL (failure condition). Additional details are deferred to Appendix B.1.

2. BinaryMechanism-CD (Algorithm 5): This subrou-

tine is used to privately count the sum of the difference in the count of distinct elements between consecutive timesteps. The mechanism is an extension of the Binary Mechanism (Chan et al., 2011; Dwork et al., 2010); however, the major differences are that it uses Gaussian noise (similar to (Jain et al., 2023)) and the input is $\{-1, 0, 1\}^T$ (as opposed to $\{0, 1\}^T$ in the original). Although the BinaryMechanism-CD algorithm is similar to prior work, its privacy analysis needs to be handled carefully in our use-case, as it is closely tied to the failure behavior of the KSET. Additional details are deferred to Appendix B.2.

COUNTING-KSET$_i$ (Algorithm 2) takes as input $x_t$ and updates the KSET data structure with $x_t$ (as long as $x_t$ is not in blocklist $\mathcal{B}$ or equal to $\perp$). If the KSET$_i$ does not fail, then COUNTING-KSET$_i$ updates BinaryMechanism-CD$_i$ with the difference of the distinct sample size at time $t$ and the distinct sample size at the last timestep before $t_{last} < t$ (Line 26) that the KSET did not fail. Then BinaryMechanism-CD$_i$ outputs the noisy count of distinct elements, denoted $\hat{s}_i$. If the KSET does fail, i.e., $S_i =$ NIL, then COUNTING-KSET$_i$ skips to Line 28 which returns TOO-HIGH if the KSET fails, or $\hat{s}_i$ exceeds the threshold $k$. Note that this step is crucial for proving that COUNTING-KSET$_i$ is DP, which is presented in more detail in Section 4.1. We note that COUNTING-KSET$_i$ does not take as input the flag $ob$, because if $ob = true$, then $\mathcal{B} = \emptyset$ in Algorithm 1, so taking $\mathcal{B}$ as input is sufficient.

Finally, Algorithm 1 maintains a stream of noisy distinct counts or TOO-HIGHs, denoted $\mathcal{S}_i$, for each of the $\log T$ instances of COUNTING-KSET$_i$ instance. These streams are used to output $\mathcal{S}_i[t] \cdot 2^i$ such that $\mathcal{S}_i[t]$ is not TOO-HIGH in Line 22, which is the private count of distinct elements in the $i$-th stream at time $t$, normalized by (the inverse of) that stream's sampling rate. This final count is output for all times $t \in [T]$.

### 3.2. Main Results

In this subsection, we present our main results: Corollary 3.1 and Corollary 3.2, which summarize our main results on the privacy (Theorem 4.1), accuracy (Theorem 4.5), and space (Theorem 4.7) of Algorithm 1. Corollary 3.1 first summarizes our main results when there is a promised upper bound on the occurrence of the input stream.

**Corollary 3.1.** *For all $\eta > 0$ and $\beta \in (0, 1)$, and a stream of length $T$, universe size $|\mathcal{U}| = poly(T)$, and promised occurrence $\leq W$, there exists a $\beta$-approximate $\rho$-zCDP algorithm in the turnstile model under continual release that, with probability at least $1 - 2\beta$, outputs a $(1 \pm \eta, \max\{O(\gamma/\eta), O(\lambda/\eta^2)\})$-approximation to the number of distinct elements using space $O(\sqrt{W} \cdot polylog(T/\beta)) \cdot$*

$poly(\frac{1}{\rho\eta})$ *where* $\gamma = O\left(\sqrt{\frac{W(\log T)^3 \log(\log T/\beta)}{\rho}}\right)$ *and* $\lambda = O(\log(\log(T)/\beta))$.

The next result (Corollary 3.2) summarizes our theoretical guarantees when the stream has unbounded occurrency.

**Corollary 3.2.** *For all $\eta > 0$ and $\beta \in (0,1)$, and a stream of length $T$ and universe size $|\mathcal{U}| = poly(T)$, there exists a $2\beta$-approximate $\rho$-zCDP algorithm in the turnstile model under continual release that, with probability at least $1 - 2\beta$ outputs a $(1 \pm \eta, \max\{O(\gamma/\eta), O(\lambda/\eta^2)\})$-approximation to the number of distinct elements using space $O(T^{1/3} \cdot polylog(T/\beta)) \cdot poly(\frac{1}{\rho\eta})$ where $\gamma = O\left(T^{1/3}\sqrt{\frac{(\log T)^3 \log(\log T/\beta)}{\rho}}\right)$ and $\lambda = O(\log(\log(T)/\beta))$.*

## 4. Analysis of **CountDistinct**

This section presents details of the privacy (Theorem 4.1), accuracy (Theorem 4.5) and space (Theorem 4.7) guarantees of Algorithm 1. Omitted proofs are in Appendix D.

### 4.1. Privacy

First we present the privacy analysis, showing that Count-Distinct (Algorithm 1) is differentially private. One key challenge in the privacy analysis is to ensure that the output of the KSET data structure does not leak privacy, even in the event of its failure (i.e., if more than $k$ distinct elements are stored). Note that if the KSET *never* failed, then one could simply sum up the difference in the output sizes of the KSET over consecutive timesteps, i.e., the stream $s_x \in \{-1, 0, 1\}^T$ using BinaryMechanism-CD. Assuming no failures, the sensitivity of $s_x$ can be bounded in terms of the maximum occurrency $W$, since in this case the KSET returns exact counts. However, the failure of KSET cannot be avoided or absorbed into the $\delta$ parameter because the KSET will fail with probability 1 when its capacity exceeds $k$ (see Lemma B.1).

We address this challenge by modifying the original KSET algorithm to have well-behaved failures. More precisely, we introduce a thresholding step where our algorithm returns TOO-HIGH if the KSET fails or approaches a regime where failing is a likely event. The latter can be estimated privately by verifying whether the binary tree output is too large, $\hat{s}_x > \tau$, and returning TOO-HIGH if so (see COUNTING-KSET, Algorithm 2).

By doing so, we can make a coupling argument between our algorithm and a much simpler algorithm (COUNTING-DICT, Algorithm 3) that simply stores the exact counts of elements and computes $\hat{s}_x$ via BinaryMechanism-CD and has the same thresholding step: if $\hat{s}_x > \tau$, return

TOO-HIGH. Note that COUNTING-DICT is not space-efficient, and we only introduce it for analysis purposes. In COUNTING-DICT, the sensitivity of $s_x$ can be bounded in terms of the maximum occurrency $W$ for all timesteps. We then use the coupling argument to bound the sensitivity of $s_x$ in COUNTING-KSET for all timesteps (barring specific bad events whose failure probability is negligible and absorbed into the DP failure probability).

---

**Algorithm 3** COUNTING-DICT

**Require:** Stream $x_1, \ldots, x_T \in \mathcal{U}$, relative error $\eta \in (0, 0.5)$, privacy parameter $\rho$, failure probability $\beta$, occurrency bound $W$ on elements, substream index $i$, list of blocklisted elements $\mathcal{B}$

1: Let $\hat{s}_i = 0$ and $\tau = 16\max\{\gamma/\eta, 32\lambda/\eta^2\} + 2\sqrt{2}\frac{(\log T+1)^{3/2}\sqrt{W\log(20T\lceil\log T\rceil/\beta)}}{\sqrt{\rho}}$

2: Initialize BinaryMechanism-CD$_i$ with parameters: privacy parameter $\rho/L$ and occurrency bound $W$

3: Initialize DICT$_i$ dictionary data structure of size $|\mathcal{U}| \times T$

4: Initialize $F_{i,last} = 0$

5: **Update**$(x_t, \mathcal{B})$:

6: **for** update $x_t$ **do**

7:     **if** $x_t \neq \perp$ and $x_t \notin \mathcal{B}$ **then**

8:         **if** $x_t$ is an insertion **then**

9:             DICT$_i[x_t][t]$ = DICT$_i[x_t][t-1] + 1$

10:         **else**

11:             DICT$_i[x_t][t]$ = DICT$_i[x_t][t-1] - 1$

12:         **end if**

13:         Let $s_i \leftarrow \sum_{u \in \mathcal{U}} \mathbf{1}_{\text{DICT}_i[u][t]>0}$

14:         $\hat{s}_i \leftarrow$ BinaryMechanism-CD$_i$.Update$(s_i - F_{i,last})$

15:         Save $F_{i,last} \leftarrow s_i$

16:     **else**

17:         $\hat{s}_i \leftarrow$ BinaryMechanism-CD$_i$.Update$(0)$

18:     **end if**

19:     **if** $\hat{s}_i > \tau$ **then**

20:         Return TOO-HIGH

21:     **else**

22:         Return $\hat{s}_i$

23:     **end if**

24: **end for**

---

We present the main theorem of the privacy guarantee below. The proof requires showing that the output stream published by COUNTING-KSET$_i$ is DP (see Corollary 4.4), which is argued by showing that the outputs of COUNTING-KSET$_i$ and COUNTING-DICT$_i$ are identical except with probability at most $\beta$ (see Lemma 4.2). Since all the operations after calling the subroutine COUNTING-KSET$_i$ in Lines 14 and 19 of Algorithm 1 is post-processing, we will have shown that CountDistinct is approximate zCDP (see Definition 2.3) in Theorem 4.1.

**Theorem 4.1.** *CountDistinct (Algorithm 1) is*

*1. $\beta$-approximate $\rho$-zCDP if $ob = true$,*

*2. $2\beta$-approximate $\rho$-zCDP if $ob = false$.*

To prove Theorem 4.1, we first establish that the outputs of COUNTING-KSET$_i$ and COUNTING-DICT$_i$ are identical with high probability (Lemma 4.2, proven in Appendix D.1). Define CountDistinct' as a variant of CountDistinct that replaces calls to COUNTING-KSET$_i$ (Algorithm 2) in Line 9, Line 14, Line 19 with calls to COUNTING-DICT$_i$ (Algorithm 3) for $i \in [L]$.

**Lemma 4.2.** *Fix the randomness used across runs of Count-Distinct and CountDistinct'. Fix $i \in [L]$, and let $K$ and $E$ denote the output distributions of COUNTING-KSET$_i$ and COUNTING-DICT$_i$ respectively. If $k \geq \tau + O\left(\frac{polylog(T/\beta)\sqrt{W}}{\sqrt{\rho}}\right)$, then the total variation distance of the two distributions, $d_{TV}(K, E) \leq \beta/L$.*

To prove that COUNTING-KSET$_i$ is differentially private (Corollary 4.4), we first show that COUNTING-DICT$_i$ satisfies differential privacy (Lemma 4.3).

**Lemma 4.3.** *COUNTING-DICT$_i$ (Algorithm 3) is $\rho/L$-zCDP, if $ob = true$, and is $\beta/L$-approximate $\rho/L$-zCDP, if $ob = false$.*

The full proof of Lemma 4.3 is in Appendix D.2; we provide a proof sketch here to highlight the key ideas. Recall that COUNTING-DICT$_i$ uses a dictionary data structure DICT to store the counts of the elements seen in the stream, and the difference in the number of distinct elements is exactly computed from DICT and fed as input to BM-Count-Distinct$_i$, which outputs a noisy distinct element count $\hat{s}_i$. Thus, in order to prove the privacy guarantee of COUNTING-DICT$_i$, we need to show that BM-Count-Distinct$_i$ is DP. Then the output of COUNTING-DICT$_i$ will simply be post-processing on the DP output of BM-Count-Distinct$_i$ (either $\hat{s}_i$ or TOO-HIGH), so it will also be DP. Hence, we prove that BM-Count-Distinct$_i$ when used in COUNTING-DICT$_i$ is $\beta/L$-approximate $\rho/L$-zCDP (in Lemma D.4), which implies Lemma 4.3.

We next sketch the proof of Lemma D.4, that COUNTING-DICT$_i$ is $\rho/L$-zCDP. On a high-level, we need to argue that if $x$ and $x'$ are neighboring streams, then the resulting streams (after hashing and blocklisting, see Definition D.2) that are fed as input to BM-Count-Distinct$_i$ can differ in at most $W + 1$ positions with probability $1 - \beta/L$, for occurrency bound $W = T^{2/3}$ (see Lemma D.6). Opening up the analysis of the binary tree mechanism in BM-Count-Distinct$_i$, we show that the sensitivity of the nodes over all levels of the binary tree is at most $2\sqrt{(W + 1)(\log(T) + 1)}$. Thus adding Gaussian noise proportional to this quantity to each node of the binary tree preserves $\rho/L$-zCDP with probability $\beta/L$.

We emphasize that the privacy argument for the BM-Count-Distinct$_i$ instance in COUNTING-DICT$_i$ cannot be directly applied to the BM-Count-Distinct$_i$ instance in COUNTING-KSET$_i$. This is because in COUNTING-KSET$_i$, the output of the KSET is used to compute the input stream to BM-Count-Distinct$_i$, and failures of the KSET can lead to a large difference in the outputs of BM-Count-Distinct$_i$ on neighboring streams. Thus the failure behavior of the KSET must be handled carefully in the privacy analysis. Corollary 4.4 gives the resulting privacy guarantee for COUNTING-KSET$_i$.

**Corollary 4.4.** *COUNTING-KSET$_i$ (Algorithm 2) is (1) $(\beta/L)$-approximate $(\rho/L)$-zCDP if $ob = true$, and (2) $(2\beta/L)$-approximate $(\rho/L)$-zCDP if $ob = false$.*

*Proof.* From Lemma 4.3 we know that COUNTING-DICT$_i$ is $\beta/L$-approximate $\rho/L$-zCDP. In particular, for $ob = true$, COUNTING-DICT$_i$ is $\rho/L$-zCDP. From Lemma 4.2, we know that the output distribution of COUNTING-DICT$_i$ and COUNTING-KSET$_i$ is identical except with probability $\beta/L$. The claims for when $ob = true$ vs $ob = false$ follows. □

We are finally ready to prove Theorem 4.1.

*Proof of Theorem 4.1.* We first argue about the case when $ob = false$ as this is the more general case.

The randomness of CountDistinct can be viewed as a joint probability distribution $\mathcal{R}_{CD} = \mathcal{R}_g \times \mathcal{R}_{BL} \times \mathcal{R}_{KC_1} \times \ldots \times \mathcal{R}_{KC_L}$ where $\mathcal{R}_g$ denotes the randomness from picking hash function $g$ (in Line 7 of Algorithm 1), $\mathcal{R}_{BL}$ denotes the randomness associated with sampling an element to add to blocklist $\mathcal{B}$ (in Line 16), and $\mathcal{R}_{KC_i}$ denotes the randomness from the subroutine COUNTING-KSET$_i$ for $i \in [L]$. Similarly, the randomness of CountDistinct' can be viewed as a joint probability distribution $\mathcal{R}_{CD'} = \mathcal{R}_g \times \mathcal{R}_{BL} \times \mathcal{R}_{EC_1} \times \ldots \times \mathcal{R}_{EC_L}$ where $\mathcal{R}_g$ denotes the randomness from picking a hash function $g$, $\mathcal{R}_{BL}$ denotes the randomness associated with sampling an element to add to blocklist $\mathcal{B}$ and $\mathcal{R}_{EC_i}$ denotes the randomness from the subroutine COUNTING-DICT$_i$ for $i \in [L]$.

We first define an identity coupling over the randomness $\mathcal{R}_g$ of picking the hash function and the randomness $\mathcal{R}_{BL}$ of blocklisting between CountDistinct and CountDistinct'. In other words, we fix the same hash function $g$ and the same sampling rate to blocklist an item for CountDistinct and CountDistinct'. Applying Lemma 4.2, we have that the outputs of COUNTING-KSET$_i$ and COUNTING-DICT$_i$ are identical except with probability $\beta/L$. In particular, from Corollary 4.4, we have that COUNTING-KSET$_i$ is $2\beta/L$-approximate $\rho/L$-zCDP. Since we have a total of $L$ substreams, using a union bound argument, the entire

CountDistinct algorithm is $2\beta$-approximate $\rho$-zCDP by basic composition of zCDP (Theorem A.1).[5]

In the case when $ob = true$, note that we only have to consider the identity coupling of over hashing items using hash function $g$ in CountDistinct and CountDistinct'. The claim that the outputs are identical except with probability $\beta/L$ follows from Lemma 4.2. But now, from Corollary 4.4, we have that COUNTING-KSET$_i$ is $\beta/L$-approximate $\rho/L$-zCDP, and the rest of the argument follows from a union bound and composition. $\square$

## 4.2. Accuracy

We present our main accuracy theorem in Theorem 4.5. Our accuracy theorem is also parameterized by the value of the boolean flag $ob$ and gives different error/space trade-offs according to whether $ob$ is true or false. In particular if $ob$ is true, meaning our algorithm is promised that the occurrency of input streams is bounded by $W$, then we get additive error only that has $\sqrt{W}$ dependency. If $ob$ is false, then our algorithm makes no assumption on the occurrency of the input stream and therefore incurs a higher additive error. Omitted accuracy proofs are in Appendices D.3 and D.4.

**Theorem 4.5** (Accuracy of Algorithm 1). *Let $F(t)$ be the correct number of distinct elements of the stream at time $t$ and let $\lambda = 2\log(40\lceil\log(T)\rceil/\beta)$. When ob=true, let $\gamma = \sqrt{\frac{4(W+1)(\log T+1)^3\log(10(\log T+1)/\beta)}{\rho}}$; when ob=false, let $\gamma = \sqrt{\frac{4(T^{2/3}+1)(\log T+1)^3\log(10(\log T+1)/\beta)}{\rho}} + 3T^{1/3}\log(T^{1/3}\lceil\log T\rceil/\beta)$. For a fixed timestep $t \in [T]$, with probability at least $1-2\beta$, the output of Algorithm 1 at time $t$ is a $(1\pm 4\eta, 32\max\{\gamma/\eta, 32\lambda/\eta^2\})$-approximation of $F(t)$ for any $\eta \in (0, 0.5)$.*

To prove Theorem 4.5, we use three helper lemmas, all proved in Appendix D.4. Lemma D.7 bounds the number of elements in the substream after hashing. Lemma D.8 proves the accuracy of BinaryMechanism-CD algorithm. Lemma D.9 bounds the size of the blocklist when $ob$ is false. With the help of these lemmas, we can show the accuracy of COUNTING-DICT, as an intermediate step in the analysis. The proof of Theorem 4.6 is deferred to Appendix D.3.

**Theorem 4.6.** *Let $F(t)$ be the correct number of distinct elements of the stream at time $t$ and let $\lambda = 2\log(40\lceil\log(T)\rceil/\beta)$. When ob is true, let $\gamma = \sqrt{\frac{4(W+1)(\log T+1)^3\log(10(\log T+1)/\beta)}{\rho}}$ and when ob is false, let $\gamma = \sqrt{\frac{4(T^{2/3}+1)(\log T+1)^3\log(10(\log T+1)/\beta)}{\rho}} + 3T^{1/3}\log(T^{1/3}\lceil\log T\rceil/\beta)$. For a fixed timestep $t \in [T]$, with probability at least $1-\beta$, the output of Algorithm 1 at time $t$ with COUNTING-DICT$_i$ as the subroutine is a*

$(1\pm 4\eta, 32\max\{\gamma/\eta, 32\lambda/\eta^2\})$-*approximation of $F(t)$ for any $\eta \in (0, 0.5)$.*

With this, we can finally prove Theorem 4.5, by showing that substituting COUNTING-KSET in place of COUNTING-DICT still allows high accuracy of Algorithm 1.

*Proof of Theorem 4.5.* We apply a union bound argument that combines Lemma 4.2 and Theorem 4.6. By Lemma 4.2, we can link the accuracy of COUNTING-DICT$_i$ to that of COUNTING-KSET$_i$ and with probability at least $1-\beta$, the output distributions of all the $L$ instances of COUNTING-DICT$_i$ and COUNTING-KSET$_i$ used in Algorithm 1 are the same. Furthermore, Theorem 4.6 gives desired accuracy with probability at least $1 - \beta$ for Algorithm 1 with COUNTING-DICT$_i$. Thus, by a union bound over the two events that (1) the output of COUNTING-DICT$_i$ matches the output of COUNTING-KSET$_i$ for $i \in [L]$ and (2) COUNTING-DICT$_i$ is accurate. $\square$

## 4.3. Space Complexity

Finally, we present the space guarantees of our algorithm below. As with the privacy and accuracy result, Theorem 4.7 is parameterized by the value of the boolean flag $ob$. When $ob$ is true and the input stream is promised to have occurrency bounded by $W$, the space is only polynomial in $W$. When $ob$ is false, the algorithm allows general input streams, and internally enforces a bound $T^{2/3}$ on the stream's occurrency using blocklisting, which requires more space. We defer the proof of Theorem 4.7 to Appendix D.5.

**Theorem 4.7.** *With probability at least $1 - \beta$, assuming the universe size $|\mathcal{U}| = poly(T)$: If ob is true, the space complexity of Algorithm 1 is $O(\sqrt{W} \cdot polylog(T/\beta)) \cdot poly(\frac{1}{\rho\eta})$. If ob is false, the space complexity of Algorithm 1 is $O(T^{1/3} \cdot polylog(T/\beta)) \cdot poly(\frac{1}{\rho\eta})$.*

## 5. Conclusions

In this paper we designed the first space-efficient differentially private algorithms for the count distinct element problem in the turnstile model. This result addresses an open question of (Jain et al., 2023), showing that it is possible to design a low memory DP algorithm for this problem in the turnstile setting. While we show that any algorithm that uses blocklisting techniques cannot do any better in terms of space, an interesting open question is to prove unconditional space bounds for any DP continual release algorithm addressing the problem (regardless of the techniques used). The current theoretical understanding of space lower bounds in the DP streaming setting is very limited. Only recently (Dinur et al., 2023) gave the first space DP lower bound for any problem, under cryptographic assumptions; any future progress in this direction would be interesting.

---

[5]Note that Theorem A.1 gives an even tighter guarantee, but we use this slightly weaker composition for a cleaner presentation.

## Acknowledgments

R.C., T.M., and T.O. were supported in part by NSF grant CNS-1942772 and 2138834 (CAREER), a Google Cyber NYC Award, and the Center for Smart Streetscapes, an NSF Engineering Research Center, under grant agreement EEC-2133516.

## Impact Statement

This paper presents work whose goal is to advance the field of Machine Learning. There are many potential societal consequences of our work, none which we feel must be specifically highlighted here.

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

# A. Additional Tools and Background

## A.1. Additional Related Work

**DP continual release algorithms.** Differential privacy (DP) (Dwork et al., 2006) has become the *de facto* standard of private computation in algorithm design. In the context of streaming algorithms, the standard DP model is the *continual release* model, first introduced by (Dwork et al., 2010; Chan et al., 2011). In this model, algorithms should preserve privacy of the input, even if a solution is observed at each timestep (as opposed to the *one-shot* model where the adversary can obtain only *one* solution at the end of the stream). Celebrated results in the continual release model include the well-known binary mechanism for releasing sum statistics in a binary data stream with $O(\varepsilon^{-1}\log^2(T))$ additive error (Dwork et al., 2010; Chan et al., 2011). Significant work has expanded on this foundational result in various directions, including handling non-binary streams (Thakurta & Smith, 2013; Fichtenberger et al., 2021), sliding windows (Bolot et al., 2013), and improving the space/utility tradeoffs (Dvijotham et al., 2024). The latter work also provided algorithms for counting distinct elements with additive error of $O(\log^{1.5}(T))$ in insertion-only streams. Later work (Ghazi et al., 2023) also focused on counting distinct elements in the sliding window model, achieving polylogarithmic additive error in the window size. For insertion-only streams, (Epasto et al., 2023) gave the first DP algorithms with space $O(\text{poly}\log(T))$ and a $(1+\eta)$-multiplicative and $O(\text{poly}\log(T))$ additive error for counting distinct elements and frequency moment estimation in the insertion-only and sliding window model under continual release.

**DP one-shot streaming algorithms.** A more restricted setting is the one-shot streaming model, where the analyst only seeks to output a solution at the end of the stream. In the one-shot setting, (Desfontaines et al., 2019) showed membership inference attacks for a large class of non-DP sketching algorithms for counting distinct elements, implying that they do not preserve any reasonable notion of privacy. In the private sphere and for a related problem, designing low-space DP algorithms for the general frequency moment estimation problem has been well-explored (Smith et al., 2020; Pagh & Stausholm, 2021; Blocki et al., 2012; Dickens et al., 2022; Hehir et al., 2023). Specifically (Wang et al., 2022) showed that a well-known streaming algorithm called the $\mathbb{F}_p$ sketch preserves DP as is. (Blocki et al., 2023) gave a black-box transformation for turning non-DP streaming algorithms into DP streaming algorithms while still preserving sublinear space and accuracy guarantees. These problems have also been explored in the *pan-privacy* streaming model, where DP is preserved even if the internal memory of the algorithm is compromised (Dwork et al., 2010; Mir et al., 2011).

## A.2. Additional Background on Differential Privacy

The privacy parameters of zCDP compose, similar to the composition guarantees of DP. Additionally, it is possible to translate between the guarantees of zCDP and DP.

**Theorem A.1** (Composition (Bun & Steinke, 2016)). *Let $\mathcal{A}$ be a $\delta$-approximate $\rho$-zCDP algorithm and $\mathcal{A}'$ be a $\delta'$-approximate $\rho'$-zCDP algorithm. Then the composition $\mathcal{A}''(x) = (\mathcal{A}(x), \mathcal{A}'(x))$ satisfies $(\delta + \delta' - \delta \cdot \delta')$-approximate $(\rho + \rho')$-zCDP.*

**Theorem A.2** (Relationship to DP (Bun & Steinke, 2016)). *For all $\rho, \delta > 0$, if algorithm $\mathcal{A}$ is $\rho$-zCDP then $\mathcal{A}$ is $(\rho + 2\sqrt{\rho\log(1/\delta)}, \delta)$-DP. Conversely, if $\mathcal{A}$ is $\varepsilon$-DP then $\mathcal{A}$ is $(\varepsilon^2/2)$-zCDP. If algorithm $\mathcal{A}$ is $\delta$-approximate $\rho$-zCDP then $\mathcal{A}$ is $(\varepsilon, \delta + (1-\delta)\delta')$-DP for all $\varepsilon \geq \rho$, where $\delta' = \exp(-(\varepsilon-\rho)^2/4\rho) \cdot \min\{1, \sqrt{\pi \cdot \rho}, \frac{1}{1+(\varepsilon-\rho)/2\rho}, \frac{2}{1+\frac{\varepsilon-\rho}{2\rho}+\sqrt{(1+\frac{\varepsilon-\rho}{2\rho})+\frac{4}{\pi\rho}}}\}$.*

*Conversely, if $\mathcal{A}$ is $(\varepsilon, \delta)$-DP then $\mathcal{A}$ is $\delta$-approximate $(\varepsilon^2/2)$-zCDP.*

Finally, we present a simple mechanism that satisfies both DP and zCDP. The Gaussian Mechanism privately answers vector-valued queries by adding Gaussian noise to the true query answer. The noise is proportional to the $\ell_2$-*sensitivity* of the function, which is the maximum change in the function's $\ell_2$-norm from changing a single element in the data.

**Definition A.3** (Sensitivity (Dwork et al., 2006)). Let $f : \mathcal{X} \to \mathbb{R}^k$ be a function. Its $\ell_2$-sensitivity is defined as

$$\max_{x \sim x' \in \mathcal{X}} \|f(x) - f(x')\|_2$$

**Theorem A.4** (Gaussian mechanism (Dwork & Roth, 2014)). *Let $f : \mathcal{X}^n \to \mathbf{R}$ be a function with $\ell_2$-sensitivity at most $\Delta_2$. Let $\mathcal{A}$ be an algorithm that on input $y$, releases a sample from $\mathcal{N}(f(y), \sigma^2)$. Then $\mathcal{A}$ is $(\Delta_2^2/(2\sigma^2))$-zCDP.*

## A.3. Concentration Bounds

We provide some basic concentration inequalities that will be used in our analyses.

**Lemma A.5** ((Bellare & Rompel, 1994)). *Let $\lambda \geq 4$ be an even integer. Let $X$ be the sum of $n$ $\lambda$-wise independent random variables which take values in $[0, 1]$. Let $\mu = E[X]$ and $A > 0$. Then,*

$$\Pr[|X - \mu| > A] \leq 8 \left( \frac{\lambda\mu + \lambda^2}{A^2} \right)^{\lambda/2} .$$

**Theorem A.6** (Multiplicative Chernoff Bound (Mitzenmacher & Upfal, 2017)). *Let $X = \sum_{i=1}^{n} X_i$ where each $X_i$ is a Bernoulli variable which takes value $1$ with probability $p_i$ and value $0$ with probability $1 - p_i$. Let $\mu = \mathbb{E}[X] = \sum_{i=1}^{n} p_i$. Then,*

1. *Upper Tail:* $\Pr[X \geq (1 + \eta) \cdot \mu] \leq \exp\left( -\frac{\eta^2 \mu}{2+\eta} \right)$ *for all $\eta > 0$;*

2. *Lower Tail:* $\Pr[X \leq (1 - \eta) \cdot \mu] \leq \exp\left( -\frac{\eta^2 \mu}{3} \right)$ *for all $0 < \eta < 1$.*

**Lemma A.7** (Chernoff Bound of Gaussian Random Variable). *For $X \sim \mathcal{N}(0, \sigma^2)$, $\Pr[|X| > t] \leq 2 \exp(-t^2/2\sigma^2)$.*

### A.4. Information Theory Basics

We provide some basic information theory definitions and facts that are used in Section E. In this paper, we use $\log$ to refer to the base 2 logarithm.

**Definition A.8.** The *entropy* of a random variable $X$, denoted by $H(X)$, is defined as $H(X) = \sum_x \Pr[X = x] \log(1/\Pr[X = x])$.

**Definition A.9.** The *conditional entropy* of random variable $X$ conditioned on random variable $Y$ is defined as $H(X|Y) = \mathbb{E}_y[H(X|Y = y)] = \sum_y \Pr[Y = y] \cdot H(X|Y = y)$.

**Definition A.10.** The *mutual information* between two random variables $X$ and $Y$ is defined as $I(X; Y) = H(X) - H(X|Y) = H(Y) - H(Y|X)$.

**Definition A.11.** The *conditional mutual information* between $X$ and $Y$ given $Z$ is defined as $I(X; Y|Z) = \mathbb{E}_z[I(X; Y|Z = z)]$

**Fact A.12.** Let $X, Y, Z$ be three random variables.

1. $H(X|Y) \geq H(X|Y, Z)$.

2. $H(X) \leq \log |\mathrm{supp}(X)|$.

3. $I(X; Y|Z) \leq H(X|Z)$.

4. Data processing inequality: for a deterministic function $f(X)$, $I(X; Y|Z) \geq I(f(X); Y|Z)$.

5. $I(X; Y|Z) \geq 0$.

## B. KSET and Binary Mechanism

### B.1. KSET Data Structure

In this subsection, we present the KSET data structure, first introduced in (Ganguly, 2007). In the non-DP turnstile setting, one key strategy for solving a variety of problems including estimating the number of distinct elements in sublinear space is to use space-efficient and dynamic *distinct sample* data structures. A distinct sample of a stream with sampling probability $p$ is a set of items such that each of the *distinct* items in the stream has an equal and independent probability of $p$ of being included in the set. (Ganguly, 2007) introduced the KSET data structure for this problem, which can be used to (non-privately) give a $(1 + \alpha)$-approximation of the number of distinct elements in a stream using roughly $O(\frac{1}{\alpha^2}(\log(T) + \log(|\mathcal{U}|) \log(|\mathcal{U}|))$ space. Our implementation of KSET is largely similar to the original version in (Ganguly, 2007), with one small change that is necessary for our privacy analysis.

Our primary reason for using the KSET data structure in the DP setting is that it allows us to maintain a distinct sample. This in turn helps us add a smaller amount of noise (proportional to the bounded occurrency of data items) to the final distinct

elements estimate released through a binary mechanism. However, the privacy analysis for the output of the KSET is quite involved and discussed in Section 4.1. To the best of our knowledge, a distinct sample data structure has not previously been used in the DP setting.

The $k$-set structure (which we denote as KSET and is formally described in Algorithm 4) is a dictionary data structure that supports insertion and deletion of data items and, with high probability, either returns the set of items $S$ present in the dictionary as long as $|S| \leq k$, or otherwise returns NIL. The structure is represented as a 2D array $H[R \times B]$ which consists of $R$ hash tables, each containing $B$ buckets, where $R = \lceil \log \frac{k}{\beta} \rceil$ and $B = 2k$. For each $r \in [R], b \in [B]$, the bucket $H[r, b]$ contains a TESTSINGLETON data structure, which tests whether or not the bucket $H[r, b]$ contains a single universe element. Details on the implementation of the TESTSINGLETON data structure are deferred to Appendix C. The $r$-th hash table in $H[R \times B]$ uses a pairwise independent hash function $h_r : \mathcal{U} \to [B]$. Upon the arrival of an update $x_t$, the KSET structure contains two main operations:

1. The Update($x_t$) operation first increments (resp., decrements) the total number $m$ of data items in the structure (which is initialized to zero) if the update is an insertion (resp., deletion) of an item. Next, for every hash table $r \in [R]$, we update the corresponding TESTSINGLETON structure in the bucket $H[r, h_r(x_t)]$.

2. The ReturnSet() operation, for every $r \in [R]$, iterates over the buckets $b \in [B]$, and checks whether the entry in the hash table $H[r, b]$ is a SINGLETON. If so, then it retrieves the data item along with its frequency and keeps track of the set of elements ($S$) as well as the total sum of frequencies of items ($m_s$) in $S$. If $m_s = m$ and $|S| \leq k$, then it returns $S$. Otherwise, the function returns NIL. We note that the latter check for $|S| \leq k$ is not included in the original version of KSET from (Ganguly, 2007), but it is crucial for our privacy analysis. We include this check to ensure that the KSET returns NIL with probability 1 in the event that there are more than $k$ elements.

Next, we state some simple properties regarding the accuracy of the KSET structure in Lemma B.1 that are used in the analysis of Algorithm 1.

**Lemma B.1** (KSET properties). *Consider a KSET data structure with capacity $k$ and failure probability $\beta$. Then, the following holds:*

1. *If strictly more than $k$ distict elements are present in the KSET, then with probability 1, KSET.ReturnSet = NIL.*

2. *If less than or equal to $k$ distict elements are present in the KSET then with probability $\geq 1 - \beta$, KSET.ReturnSet = S where $S$ is the entire set of items present in the KSET.*

3. *The space complexity of KSET is $O(k(\log T + \log |\mathcal{U}|) \log \frac{k}{\beta})$.*

*Proof.* For Item 1, from (Ganguly, 2007), we have that if $m_s = m$ then the set of retrieved items $S$ is exactly the set of distinct elements with probability 1. Observe that if there are more than $k$ elements, then either $m_s \neq m$, or the number of distinct elements returned is $|S| > k$. In both cases, the condition in Algorithm 4 fails and the output is NIL.

The proofs for Item 2 and Item 3 are identical to the one given in (Ganguly, 2007). □

### B.2. Binary Mechanism for the Count Distinct Problem

In this subsection, we present a subroutine for COUNTING-KSET (Algorithm 2) that is a modified Binary Mechanism called BinaryMechanism-CD (see Algorithm 5). This subroutine is used to compute the summation stream $s_x \in \{-1, 0, 1\}^T$ representing the difference in the number of distinct elements at timesteps $t - 1$ and $t$ which is computed from the output of the KSET in Line 14 of Algorithm 2. The algorithm BinaryMechanism-CD injects Gaussian noise (similar to (Jain et al., 2023)) — as opposed to Laplace noise used in the original versions of the Binary Mechanism (Chan et al., 2011; Dwork et al., 2010) — proportional to the sensitivity of the summation stream.

We first note that if the input to BinaryMechanism-CD represented the *exact* difference in the number of distinct elements over consecutive timesteps, then the sensitivity of the summation stream in terms of occurrency can be calculated in a straightforward manner using arguments similar to (Jain et al., 2023). However, in for our use of BinaryMechanism-CD in the algorithm COUNTING-KSET, BinaryMechanism-CD receives as input the difference in the number of distinct elements from the output of the KSET. Importantly, the failure behavior of the KSET needs to be accounted for when

---
**Algorithm 4** KSET data structure (Ganguly, 2007)
---
**Require:** Capacity parameter $k$, failure probability $\beta$
 1: Initialize 2D array $H[\log \frac{k}{\beta} \times 2k]$
 2: $R \leftarrow \lceil \log \frac{k}{\beta} \rceil$, $B \leftarrow 2k$
 3: Let $h_r : \mathcal{U} \rightarrow [B]$ be a pairwise independent function for $r = 1, \dots, R$
 4: Initialize $m = 0$
 5: **Update**($x_t$)**:** {Process update $x_t$}
 6: **if** $x_t$ is an insertion **then**
 7:     $m \leftarrow m + 1$
 8: **else if** $x_t$ is a deletion **then**
 9:     $m \leftarrow m - 1$
10: **end if**
11: **for** $r \in [R]$ **do**
12:     $H[r, h_r(x_t)]$.TSUPDATE($x_t$) {see Algorithm 6}
13: **end for**
14: **ReturnSet():**
15: Initialize set $S = \{\}$, $m_s = 0$
16: **for** $r \in [R]$ **do**
17:     **for** $b \in [B]$ **do**
18:         **if** $H[r, b]$.TSCARD()[0] == SINGLETON **then**
19:             $(x, c) \leftarrow H[r, b]$.TSCARD()[1], $H[r, b]$.TSCARD()[2] {see Algorithm 6}
20:             Insert $(x, f_x)$ to $S$
21:             $m_s \leftarrow m_s + f_x$
22:         **end if**
23:     **end for**
24: **end for**
25: **if** $m_s = m$ and $|S| \leq k$ **then**
26:     Return $S$
27: **else**
28:     Return NIL
29: **end if**

---
**Algorithm 5** BinaryMechanism-CD
---
**Require:** Count distinct summation stream $y_1, y_2, \dots, y_T \in \{-1, 0, 1\}^T$, privacy parameter $\rho > 0$, occurrence $W > 0$
 1: Initialize each $\alpha_i = 0$ and $\hat{\alpha}_i = 0$
 2: Let $\rho' = \frac{\rho}{2(W+1)(\log T + 1)}$
 3: **Update**($y_t$)**:**
 4: **for** every update $y_t$ **do**
 5:     Express $t$ in binary from: $t = \sum_j \mathsf{Bin}_j(t) 2^j$
 6:     Let $i = \min\{j : \mathsf{Bin}_j(t) = 1\}$ be the least significant binary digit, and set $\alpha_i = \sum_{j=0}^{i-1} \alpha_j + y_t$
 7:     **for** $j = 0, 1, \dots, i - 1$ **do**
 8:         Set $\alpha_j = 0$ and $\hat{\alpha}_j = 0$
 9:     **end for**
10:     Set $\hat{\alpha}_i = \alpha_i + \mathcal{N}(0, 1/\rho')$
11:     **Return** $\mathcal{B}(t) = \sum_{j:\mathsf{Bin}_j(t)=1} \hat{\alpha}_j$
12: **end for**

---

arguing about the sensitivity of the resulting summation stream computed from the KSET output (when it does not fail). In order to do this, we use a coupling argument to show that the output stream of the algorithm COUNTING-KSET is close to the output stream of an algorithm that exactly computes the number of distinct elements and feeds the difference over consecutive timesteps to BinaryMechanism-CD as input. The privacy analysis for BinaryMechanism-CD in the latter algorithm is similar to (Jain et al., 2023) and is given in Lemma D.4. The privacy guarantee of our application of

BinaryMechanism-CD inside COUNTING-KSET is implicitly derived in the privacy analysis of COUNTING-KSET via the coupling argument of Lemma 4.2 and the claim that COUNTING-KSET is DP in Corollary 4.4.

The accuracy guarantee of BinaryMechanism-CD follows from Lemma D.8 in which we consider the overall accuracy of $L$ instances of BinaryMechanism-CD as instantiated in Line 3 of COUNTING-KSET. Finally, the space complexity of BinaryMechanism-CD is $O(\log(T))$ and this follows from (Chan et al., 2011).

## C. Additional Details on KSET

We describe the TESTSINGLETON data structure (Algorithm 6) which is a building block of the KSET data structure in more detail.

---
**Algorithm 6** TEST-SINGLETON data structure
---
**Require:** Input stream $x_1, x_2, \ldots, x_T$
 1: Initialize $m \to 0, U \to 0, V \to 0$
 2: **TSUPDATE**$(x_t)$**:**
 3: **if** $x_t$ is an insertion of item $i$ **then**
 4:     $m \leftarrow m + 1, U \to U + i, V \to V + i^2$
 5: **else if** $x_t$ is a deletion **then**
 6:     $m \leftarrow m - 1, U \to U - i, V \to V - i^2$
 7: **end if**
 8: **TSCARD():**
 9: **if** $m = 0$ **then**
10:     Return EMPTY
11: **else if** $U^2 = m \cdot V$ **then**
12:     Return (SINGLETON, $U/m$, $m$)
13: **else**
14:     Return COLLISION
15: **end if**
---

The TESTSINGLETON data structure supports the following operations:

1. An update operation, TSUPDATE$(x_t)$, which updates three counters — $m_{TS}$, $U$, and $V$ (all initialized to zero) preserving the following invariants throughout the stream:

$$m_{TS} = \sum_{a \in \mathcal{U}} f_a, \quad U = \sum_{a \in \mathcal{U}} f_a \cdot a, \quad V = \sum_{a \in \mathcal{U}} f_a \cdot a^2.$$

   More precisely, for an non-empty update $x_t$ corresponding to data item $a$, TSUPDATE$(x_t)$ performs the following update:

$$m_{TS} := m_{TS} + 1, \quad U = U + a, \quad V = V + a^2$$
$$\text{(for an addition)}$$
$$m_{TS} := m_{TS} - 1, \quad U = U - a, \quad V = V - a^2$$
$$\text{(for a deletion)}.$$

2. A check operation, TSCARD(), which determines whether the TESTSINGLETON data structure: (1) is empty, (2) contains a single element, or (3) has more than a single element. The function returns, in each case respectively: (1) EMPTY (this happens if $m_{TS} = 0$); (2) the triplet SINGLETON, the element, and its frequency (this happens if $U^2 = m_{TS} \cdot V$); or (3) COLLISION (if the last two checks fail). It is easy to see that the unique item returned (in the SINGLETON case) has identity $\frac{U}{m_{TS}}$ and has frequency $m_{TS}$.

## D. Omitted Proofs from Section 4

For ease of analysis, we define the streams produced after applying the hash function and the blocklisting procedure (in the case when $ob = false$) as follows. These streams will be used as intermediate steps in the analysis of CountDistinct to

separately reason about the hashing and blocklisting procedures, and their impact on sensitivity of the resulting streams.

**Definition D.1.** Define $\mathcal{S}_{i,g}$ as the substream of $x$ after applying hash function $g$. That is, let $a$ be the item contained in the update $x_t$. Then $\mathcal{S}_{i,g}[t] = x_t$ if $g(a) = i$ and $\mathcal{S}_{i,g}[t] = \perp$ otherwise.

**Definition D.2.** Define $\mathcal{S}_{i,B}$ as the stream of updates produced from $\mathcal{S}_{i,g}$ after checking whether the item corresponding to the update $x_t$ is in the blocklist $\mathcal{B}$ before time $t$ or not. That is, if the item corresponding to $x_t$ is in $\mathcal{B}$ before time $t$, then $\mathcal{S}_{i,B}[t] = \perp$, otherwise $\mathcal{S}_{i,B}[t] = \mathcal{S}_{i,g}[t]$.

### D.1. Proof of Lemma 4.2 and Helper Lemma

**Lemma 4.2.** *Fix the randomness used across runs of* CountDistinct *and* CountDistinct'. *Fix* $i \in [L]$, *and let* $K$ *and* $E$ *denote the output distributions of* COUNTING-KSET$_i$ *and* COUNTING-DICT$_i$ *respectively. If* $k \geq \tau + O\left(\frac{\text{polylog}(T/\beta)\sqrt{W}}{\sqrt{\rho}}\right)$, *then the total variation distance of the two distributions,* $d_{TV}(K, E) \leq \beta/L$.

*Proof.* We start with the case when $ob = false$, which is the more involved case. Consider the randomness of COUNTING-KSET$_i$ and COUNTING-DICT$_i$. Observe that because of the fixed randomness of both hashing and blocklisting, the resulting streams $\mathcal{S}_{i,B}$ (see Definition D.2) that are respectively used to update the KSET (in the case of COUNTING-KSET$_i$) and the DICT data structure (in the case of COUNTING-DICT$_i$) are identical.

Next, define the randomness of COUNTING-KSET$_i$ as $\mathcal{R}_{KC_i} = \mathcal{R}_{KS_i} \times \mathcal{R}_{BM_i}$ where $\mathcal{R}_{KS_i}$ denotes the randomness from the KSET (Algorithm 4) and $\mathcal{R}_{BM_i}$ denotes the randomness from BinaryMechanism-CD (Algorithm 5). On the other hand, the only randomness in COUNTING-DICT$_i$ is due to the randomness of BinaryMechanism-CD, i.e., $\mathcal{R}_{EC_i} = \mathcal{R}_{BM_i}$. We emphasize that because the randomness from adding items to the blocklist has been fixed across CountDistinct and CountDistinct', the blocklist $\mathcal{B}$ passed to both COUNTING-KSET$_i$ and COUNTING-DICT$_i$ are identical.

We now want to argue that the outputs of COUNTING-KSET$_i$ and COUNTING-DICT$_i$ are the same except with probability $\beta$. Let the BinaryMechanism-CD instance in COUNTING-KSET$_i$ be denoted as BinaryMechanism-CD$_{KC_i}$ and the BinaryMechanism-CD instance in COUNTING-DICT$_i$ as BinaryMechanism-CD$_{EC_i}$, and fix the randomness used in BinaryMechanism-CD$_{KC_i}$ and BinaryMechanism-CD$_{EC_i}$.

We claim that there are two bad events for which the outputs of COUNTING-KSET$_i$ and COUNTING-DICT$_i$ may differ.

- **E$_1$**: There exists a timestep where the true count $\leq k$ and the KSET outputs NIL.

- **E$_2$**: There exists a timestep where the true count is $> k$ and the noisy count of COUNTING-DICT$_i \leq k$.

By setting $k \geq \tau + O\left(\frac{\text{polylog}(T/\beta)\sqrt{W}}{\sqrt{\rho}}\right)$, we argue that both events happen with probability at most $\beta$.

For the event **E$_1$**, the probability of the KSET outputting NIL happens with probability at most $\beta/2TL$ for one timestep. This is because, in Line 4 of COUNTING-KSET$_i$, we set $k$ as the capacity of the KSET and $\beta/2TL$ as the failure probability. So by Item 2 in Lemma B.1 and union bound over all timesteps, this event happens with probability at most $\beta/2L$.

For the event **E$_2$**, Lemma D.3 (below) bounds the probability of **E$_2$** as $\beta/2L$ over all timesteps for our choice of $k$.

Now, conditioned on bad events **E$_1$** and **E$_2$** *not* occurring over all timesteps, we argue that the outputs of COUNTING-KSET$_i$ and COUNTING-DICT$_i$ are identical. Let $t_1$ be the *first* timestep that the true count is $> k$. Let $t_2 > t_1$ be the next timestep that the true count is $\leq k$. We next consider the outputs of the two algorithms by cases across timesteps.

- **Case 1:** $1 \leq t < t_1$**.** Conditioned on event **E$_1$** not occurring, the KSET does not output NIL during this time epoch, which means that the inputs to BinaryMechanism-CD$_{KC_i}$ and BinaryMechanism-CD$_{EC_i}$ are identical, since both BinaryMechanism-CD$_{KC_i}$ and BinaryMechanism-CD$_{EC_i}$ will be updated with only the update from the previous timestep. In this case, the resulting noisy outputs will be the same under the fixed randomness, and the output of COUNTING-KSET$_i$ and COUNTING-DICT$_i$ after the thresholding step (comparison to $\tau$) is identical.

- **Case 2:** $t_1 \leq t < t_2$**.** For timesteps in this epoch, both COUNTING-KSET$_i$ and COUNTING-DICT$_i$ will output TOO-HIGH. Since the true count is $> k$ for all $t_1 \leq t < t_2$, and conditioned on **E$_2$** not occurring, the output of COUNTING-DICT$_i$ must be TOO-HIGH. Also by Item 1 in Lemma B.1, the KSET outputs NIL for this time period with probability 1, which means that the output of COUNTING-KSET$_i$ is also TOO-HIGH.

- **Case 3:** $t = t_2$. Next we argue about the output of COUNTING-DICT$_i$ and COUNTING-KSET$_i$ at timestep $t_2$ when the true count $\leq k$. Conditioning on event $\mathbf{E}_1$ not occurring, the KSET does not output NIL at timestep $t_2$ because the true count $\leq k$. Moreover, observe that BinaryMechanism-CD$_{KC_i}$ is not updated over $t_1 < t < t_2$ and is only updated at timestep $t_2$ because the KSET does not output NIL. Also, by construction of COUNTING-KSET$_i$, we claim that BinaryMechanism-CD$_{KC_i}$ is fed a sequence of inputs $(+1, -1, 0)$ at timestep $t_2$ that result in the same sum and the same length as in BinaryMechanism-CD$_{EC_i}$ over $t_2 - t_1$ timesteps. This is because by definition, diff is the difference in the number of distinct elements between times $t_1 - 1$ and $t_2$, and $|\text{diff}| \leq t_{\text{diff}}$, as there can only be $\leq t_{\text{diff}}$ many distinct elements added (or removed) over $t_2 - t_1$. Since the length and sum of the sequence of inputs to both BinaryMechanism-CD$_{KC_i}$ and BinaryMechanism-CD$_{EC_i}$ at timestep $t_2$ is the same, the outputs of both COUNTING-KSET$_i$ and COUNTING-DICT$_i$ are the same at timestep $t_2$ under the fixed randomness.

This argument can be extended over all timesteps by iteratively considering the *next* timestep when the true count is $> k$ and the following timestep when the true count is $\leq k$. Thus when $ob = true$, except with probability $\beta/L$ corresponding to the events $\mathbf{E}_1$ and $\mathbf{E}_2$ occurring, COUNTING-KSET$_i$ and COUNTING-DICT$_i$ will produce identical outputs at each timestep. That is, the distributions of COUNTING-KSET$_i$ and COUNTING-DICT$_i$, denoted $K$ and $E$ respectively, will agree on all outcomes except a subset of probability mass $\beta/L$, which implies that that $d_{TV}(K, E) \leq \beta/L$.

For the case when $ob = true$, the blocklisting step is not needed. Then the fixed randomness between CountDistinct and CountDistinct' means that the randomness of the hash functions of both algorithms will be the same, so the resulting stream $\mathcal{S}_{i,g}$ (see Definition D.1) that is used to update the KSET (in the case of COUNTING-KSET$_i$) and the DICT data structure (in the case of COUNTING-DICT$_i$) is identical. The rest of the argument follows symmetrically to the case when $ob = false$. $\qquad\square$

### D.1.1. HELPER LEMMA

**Lemma D.3.** *Let $\mathbf{E}_{BM}$ be the event that there exists a timestep where the noisy count of BinaryMechanism-CD$_{EC_i}$ is greater than $k$ when the true count is less than $\tau$, or the noisy count of BinaryMechanism-CD$_{EC_i}$ is less than $\tau$ when the true count is greater than $k$. The probability of $\mathbf{E}_{BM}$ is at most $\beta/2L$ over all timesteps when $k \geq \tau + 2\sqrt{2}\frac{(\log T+1)^{3/2}\sqrt{W\log(4T\lceil\log(T)\rceil/\beta)}}{\sqrt{\rho}}$.*

*Proof.* For notational convenience, let $\Delta = k - \tau$. From Algorithm 5, we know that the noise that we apply to the true count is a summation of at most $m \leq \log T + 1$ Gaussian random variables, each sampled from $\mathcal{N}(0, 4W(\log T + 1)L/\rho)$. (Recall that the input privacy parameter to the binary mechanism is $\rho/L$.) Thus the overall noise added is $N(0, 4mW(\log T+1)L/\rho)$. Now we want to bound the probability that $|\mathcal{N}(0, 4mW(\log T + 1)L/\rho)| > \Delta$, which is an upper bound on the probability of $\mathbf{E}_{BM}$ occurring at a single timestep.

Applying a Chernoff bound (Lemma A.7) yields:

$$\Pr(|\mathcal{N}(0, 4mW(\log T + 1)L/\rho)| > \Delta) \leq 2\exp(-\frac{\Delta^2\rho}{8mW(\log T + 1)L})$$

We wish to bound the above term on the right by $\beta/2TL$, so that then by a union bound, the probability of $\mathbf{E}_{BM}$ over all timesteps is bounded by $\beta/2L$. This requires:

$$-\frac{\Delta^2\rho}{8mWL(\log T + 1)} \leq \log(\beta/4TL)$$

$$\iff \quad \Delta \geq 2\sqrt{2}\frac{\sqrt{mWL(\log T + 1)\log(4TL/\beta)}}{\sqrt{\rho}}$$

Recall that our goal is to set the value of $\Delta = k - \tau$ such that the above inequality always holds, and we want to set $\Delta$ to be the upper bound of the right hand side. Since $m \leq \log T + 1$ and $L = \lceil\log T\rceil \leq \log T + 1$, then, choosing $k \geq t + 2\sqrt{2}\frac{(\log T+1)^{3/2}\sqrt{W\log(4T\lceil\log(T)\rceil/\beta)}}{\sqrt{\rho}}$ will ensure that $\Delta$ is a valid upper bound, and hence that the probability of $\mathbf{E}_{BM}$ over all timesteps is bounded by $\beta/2L$. $\qquad\square$

## D.2. Proof of Lemma 4.3 and Helper Lemmas

**Lemma 4.3.** *COUNTING-DICT$_i$ (Algorithm 3) is $\rho/L$-zCDP, if $ob = true$, and is $\beta/L$-approximate $\rho/L$-zCDP, if $ob = false$.*

*Proof.* We will prove the privacy claim for the more general case when $ob = false$. Note that when $ob = true$, we do not need to deal with the failure event associated with blocklisting (Lemma D.5) and thus $\beta = 0$ and COUNTING-DICT$_i$ (Algorithm 3) is $\rho/L$-zCDP.

The key point we must show is that when neighboring streams are input to COUNTING-DICT$_i$, then the internal streams passed to BinaryMechanism-CD$_i$ inside of COUNTING-DICT$_i$ will remain neighboring. Once this is shown, then we can directly apply Lemma D.4, which shows that this instance of BinaryMechanism-CD$_i$ inside COUNTING-DICT$_i$ is differentially private. Thus we must show that even after applying the hashing and blocklisting operations to the original neighboring input streams, the resulting processed streams remain neighboring.

The randomness of CountDistinct' can be viewed as a joint probability distribution $\mathcal{R}_{CD'} = \mathcal{R}_g \times \mathcal{R}_{BL} \times \mathcal{R}_{EC_1} \times \ldots \times \mathcal{R}_{EC_L}$ where $\mathcal{R}_g$ denotes the randomness from picking a hash function $g$ (in Line 7 of Algorithm 1), $\mathcal{R}_{BL}$ denotes the randomness from blocklisting, and $\mathcal{R}_{EC_i}$ denotes the randomness from the subroutine COUNTING-DICT$_i$ for $i \in [L]$. Let $x$ and $x'$ be neighboring streams that differ only at timestep $t^*$, in which the update (either deletion or addition) in $x$ is for item $u$, and in $x'$ is $\perp$, and fix the randomness used in CountDistinct' across runs on $x$ and $x'$.

Let $\mathcal{S}_{i,g}$ and $\mathcal{S}'_{i,g}$ be the substreams of $x$ and $x'$ produced from the hash function $g$ (see Definition D.1). Then with the fixed randomness, $\mathcal{S}_{i,g}$ and $\mathcal{S}'_{i,g}$ are neighboring. To see this, observe that for all updates except those inserting or deleting $u$, $\mathcal{S}_{i,g}$ and $\mathcal{S}'_{i,g}$ are exactly the same. For updates regarding item $u$, if $u$ is hashed into substream $i$, then $\mathcal{S}_{i,g}$ and $\mathcal{S}'_{i,g}$ will differ only in time $t^*$. Otherwise $\mathcal{S}_{i,g}$ and $\mathcal{S}'_{i,g}$ will be identical. Thus, $\mathcal{S}_{i,g}$ and $\mathcal{S}'_{i,g}$ will be neighboring streams for all $i \in [L]$.

Let $\mathcal{S}_{i,B}$ and $\mathcal{S}'_{i,B}$ be the substreams of $x$ and $x'$ produced after blocklisting (see Definition D.2). Under the fixed randomness, the timesteps at which items are first blocklisted are the same for the neighboring streams, except (possibly) for item $u$. Since the updates from substreams $\mathcal{S}_{i,B}$ and $\mathcal{S}'_{i,B}$ are stored exactly as-is in DICT, and then fed to BM-Count-Distinct as input, the input streams to BinaryMechanism-CD$_i$ are indeed neighboring.

By Lemma D.4, the output $\hat{s}_i$ of BinaryMechanism-CD$_i$ is $\beta/L$-approximate $\rho/L$-zCDP. The remainders of the operations in COUNTING-DICT$_i$ – including the thresholding step to output either the numerical value of $\hat{s}_i$ or TOO-HIGH – are simply postprocessing on the private outputs $\hat{s}_i$ of BinaryMechanism-CD$_i$, which will retain the same privacy guarantee. Thus, COUNTING-DICT$_i$ is $\beta/L$-approximate $\rho/L$-zCDP as well. $\qquad\square$

### D.2.1. HELPER LEMMAS

**Lemma D.4.** *The BinaryMechanism-CD instance in COUNTING-DICT$_i$ is $\beta/L$-approximate $\rho/L$-zCDP.*

*Proof.* Consider the binary tree produced by BM-Count-Distinct$_i$ with $\log(T)$ levels. We define a vector $G_h$ of length $T/2^h$ for each level $h \in [\log(T)]$ of the binary tree as

$$G_h[j] = s_i[j \cdot 2^h] - s_i[(j-1) \cdot 2^h]$$

for all $j \in [T/2^h]$ and $s_i[t] = \sum_{u \in \mathcal{U}} \mathbf{1}_{\text{DICT}_i[u][t]>0}$ as defined in Algorithm 3 of COUNTING-DICT$_i$. Let $G = (G_0, \ldots, G_{\log(T)})$. To prove the claim, we will bound the sensitivity of the counts stored in the binary tree represented by $G$, and then show that sufficient noise is added to each count to satisfy differential privacy. Similar to the original binary tree mechanism of (Chan et al., 2011; Dwork et al., 2010), the output of BM-Count-Distinct$_i$ at timestep $t$ can be obtained from $G$ by considering the dyadic decomposition of the interval $(0, t]$ as a sum of the individual nodes composing the interval, and this output will be private by postprocessing.[6]

First, we claim that the binary tree described by $G$ plus DP noise (which we will determine) produces the output $\hat{s}_i$. To see this, first observe that $\hat{s}_i[t] = s_i[t] - s_i[t-1] + Z[t]$ where $Z[t]$ is the noise term. Also $G_0[j] = \hat{s}_i[j] - Z[j]$. The claim follows by induction over $h \in [\log(T)]$.

---

[6](Jain et al., 2023) used similar techniques to argue about the sensitivity of their binary tree mechanism. However, their argument is more straightforward as it does not have to consider the randomness from hashing or blocklisting.

Let $G$ and $G'$ be the binary tree representation of neighboring streams $x$ and $x'$ respectively. We will show that $\|G - G'\|_2 \leq 2\sqrt{(W+1)(\log(T)+1)}$ with probability $1 - \beta/L$.

Fix $h \in [\log(T)]$ and $j \in [T/2^h]$. For ease of notation, let $j_1 = (j-1) \cdot 2^h$ and $j_2 = j \cdot 2^h$. Then

$$|G_h[j] - G'_h[j]| = |s_i[j_2] - s_i[j_1] - s'_i[j_2] + s'_i[j_1]| \leq 2, \tag{1}$$

where the inequality is due to the fact that $s_i$ and $s'_i$ can differ by at most 1 at both timesteps $j_1$ and $j_2$.

Next, observe that for a fixed $h$, the intervals $(j_1, j_2]$ are disjoint, by definition. Also, for $j \in [T/2^h]$, $G_h[j] \neq G'_h[j]$ are different in at most $W + 1$ intervals with probability $1 - \beta$ (by Lemma D.6) where $W = T^{2/3}$.

Thus, with probability $1 - \beta/L$, the (squared) $\ell_2$-sensitivity of $G$ is bounded:

$$\Delta_2^2 \leq \|G - G'\|_2^2 = \sum_{h \in [\log(T)]} \sum_{j \in [T/2^h]} (G_h[j] - G'_h[j])^2 \leq (\log T + 1)(W + 1) \cdot 2^2 \tag{2}$$

By Theorem A.4, adding Gaussian noise sampled $\mathcal{N}(0, \sigma^2)$ for $\sigma^2 = \frac{\Delta_2^2 L}{2\rho}$ to each count stored in a node of the binary tree represented by $G$ will satisfy $\rho/L$-zCDP with probability $1 - \beta/L$. Plugging in the bound on $\Delta_2^2$, it is sufficient to add Gaussian noise with variance $\sigma^2 = \frac{(\log T + 1)(W+1) \cdot L}{\rho}$. In Algorithm 3, the BinaryMechanism-CD subroutine is instantiated with privacy parameter $\rho/L$, which adjusts for the extra factor of $L$.

Finally, since the output of BinaryMechanism-CD can be obtained by postprocessing the noisy nodes of $G$, the output is $\rho/L$-zCDP with probability $1 - \beta/L$. $\qquad \square$

**Lemma D.5.** *Suppose $ob = false$. With probability at least $1 - \beta/L$, the maximum occurrency of the stream produced from the blocklisting procedure (Definition D.2) is bounded by $T^{2/3}$.*

*Proof.* Recall that the probability of blocklisting any element after an appearance in the stream is $p = \frac{\log(T^{1/3}L/\beta)}{T^{2/3}}$.

For any element $x \in \mathcal{U}$, we can bound the failure probability of the blocklist to catch an element after the maximum number of occurrences:

$$\begin{aligned}
\Pr[x \notin \mathcal{B} \text{ after } T^{2/3} \text{ appearances}] &= (1 - \frac{\log(T^{1/3}L/\beta)}{T^{2/3}})^{T^{2/3}} \\
&\leq e^{-T^{2/3} \cdot \frac{\log(T^{1/3}L/\beta)}{T^{2/3}}} \\
&= e^{-\log(T^{1/3}L/\beta)} \\
&= \frac{\beta}{T^{1/3}L}
\end{aligned}$$

The inequality in the second step comes from the fact that $(1 - a) \leq e^{-a}$ for all $a \in \mathbb{R}$.

At most $T/T^{2/3} = T^{1/3}$ elements can appear $\geq T^{2/3}$ times in a stream of length $T$. Taking a union bound, the probability that *any* of these elements is not blocklisted after $T^{2/3}$ appearances is at most $\frac{\beta}{T^{1/3}L} \cdot T^{1/3} = \beta/L$. $\qquad \square$

**Lemma D.6.** *Suppose $ob = false$. Let $x$ and $x'$ be neighboring input streams and fix the randomness of CountDistinct' across runs on $x$ and $x'$. For any $i \in [L]$, $\mathcal{S}_{i,B}$ and $\mathcal{S}'_{i,B}$ differ in at most $T^{2/3} + 1$ positions with probability $1 - \beta/L$.*

*Proof.* Let neighboring streams $x$ and $x'$ differ only at timestep $t^*$, in which the update (either deletion or addition) in $x$ is for item $u$, and in $x'$ is $\perp$, and fix the randomness used in CountDistinct' across runs on $x$ and $x'$. As shown in the proof of Lemma 4.3, the substreams $\mathcal{S}_{i,g}$ and $\mathcal{S}'_{i,g}$ are neighboring and thus will also differ at timestep $t^*$ with respect to item $u$. Thus under the fixed randomness, for all timesteps $t \neq t^*$, the same items are blocklisted in $\mathcal{S}_i$ and $\mathcal{S}'_i$.

Let $\mathbf{E}_0$ be the bad event that there exists an item $v$ that appears in the hashed substream at some timestep $t \neq t^*$ and is *not* blocklisted after $T^{2/3}$ occurrences. By Lemma D.5, the probability of $\mathbf{E}_0$ is bounded by $\beta/L$. We will condition on the event $\mathbf{E}_0$ not occurring for the remainder of the proof. Note that if the items in $\mathcal{S}_{i,g}$ and $\mathcal{S}'_{i,g}$ do not appear more than $T^{2/3}$ times for all $i \in [L]$, then naturally $\mathbf{E}_0$ does not occur.

Recall that the item $u$ appears in exactly the same timesteps in $\mathcal{S}_{i,g}$ and $\mathcal{S}'_{i,g}$ for $t \neq t^*$. Suppose that the number of appearances of $u$ in those steps is $\geq T^{2/3}$. The resulting blocklisted streams $\mathcal{S}_{i,B}$ and $\mathcal{S}'_{i,B}$ can differ in at most $T^{2/3} + 1$ timesteps because the item $u$ may be blocklisted before it appears $T^{2/3}$ times, but conditioned on $\mathbf{E}_0$ not occurring, it must be blocklisted after the $T^{2/3}$-th appearance. Since $\mathcal{S}_{i,g}$ has an extra occurrence of $u$ (at timestep $t^*$) relative to $\mathcal{S}'_{i,g}$, this means that $\mathcal{S}'_{i,B}$ can differ from $\mathcal{S}'_{i,B}$ in at most $T^{2/3} + 1$ timesteps, after which both $\mathcal{S}_{i,B}$ and $\mathcal{S}'_{i,B}$ will have 0's for all future occurrences of item $u$. $\qquad\square$

### D.3. Proof of Theorem 4.6 (Accuracy)

We restate Theorem 4.6 below for convenience.

**Theorem 4.6.** *Let $F(t)$ be the correct number of distinct elements of the stream at time $t$ and let $\lambda = 2\log(40\lceil\log(T)\rceil/\beta)$. When ob is true, let $\gamma = \sqrt{\frac{4(W+1)(\log T+1)^3 \log(10(\log T+1)/\beta)}{\rho}}$ and when ob is false, let $\gamma = \sqrt{\frac{4(T^{2/3}+1)(\log T+1)^3 \log(10(\log T+1)/\beta)}{\rho}} + 3T^{1/3}\log(T^{1/3}\lceil\log T\rceil/\beta)$. For a fixed timestep $t \in [T]$, with probability at least $1 - \beta$, the output of Algorithm 1 at time $t$ with COUNTING-DICT$_i$ as the subroutine is a $(1 \pm 4\eta, 32\max\{\gamma/\eta, 32\lambda/\eta^2\})$-approximation of $F(t)$ for any $\eta \in (0, 0.5)$.*

*Proof of Theorem 4.6.* **When ob is true:** The proof relies on the following lemmas, which ensure that for a specific timestep $t$, the good events occur with high probability:

1. In all substreams $i \in [L]$, the correct number of distinct elements in the substream $i$ by hashing, denoted $F_i(t)$, is also a good estimator for the number of distinct elements in the entire stream at timestep $t$, denoted $F(t)$ (Lemma D.7). That is, for all $i \in [L]$, the following two conditions hold at the same time for any specific timestep $t$ with probability at least $1 - \beta/5$ for any $\eta \in (0, 0.5)$:

   (a) $\forall i \in [L]$ with $F(t) \geq 2^i \cdot \frac{4\lambda}{\eta^2}$, we have $(1-\eta)\frac{F(t)}{2^i} \leq F_i(t) \leq (1+\eta)\frac{F(t)}{2^i}$

   (b) $\forall i \in [L]$ with $F(t) < 2^i \cdot \frac{4\lambda}{\eta^2}$, we have $\frac{F(t)}{2^i} - \frac{4\lambda}{\eta} \leq F_i(t) \leq \frac{F(t)}{2^i} + \frac{4\lambda}{\eta}$.

2. BinaryMechanism-CD (Algorithm 5) is accurate (Lemma D.8). That is, for all $i \in [L]$, we have $|F_i(t) - \hat{s}_i(t)| \leq \gamma = \sqrt{\frac{4(W+1)(\log T+1)^3 \log(10(\log T+1)/\beta)}{\rho}}$ with probability $1 - \beta/5$.

3. For any stream $i$, if the correct number of distinct elements in the subtream $i$ is below a certain threshold then COUNTING-DICT$_i$ will not output TOO-HIGH (Lemma D.3). Plugging in $\beta/5L$ into Lemma D.3 yields that if $F_i(t) \leq 16\max\{\gamma/\eta, 32\lambda/\eta^2\}$, then COUNTING-DICT$_i$ will not output TOO-HIGH, i.e. the noisy count $\hat{s}_i \leq \tau = 16\max\{\gamma/\eta, 32\lambda/\eta^2\} + 2\sqrt{2}\frac{(\log T+1)^{3/2}\sqrt{W\log(20T\lceil\log T\rceil/\beta)}}{\sqrt{\rho}}$, with probability at least $1 - \beta/5$.

To prove the desired accuracy claim, we will condition on all three high-probability events listed above occurring at timestep $t$. Note that each of the three events occur with probability $1 - \beta/5$. Thus all three events will happen with probability at least $1 - \frac{3}{5}\beta \geq 1 - \beta$ by a union bound.

We consider two cases for the number of distinct elements of the stream at time $t$ denoted by $F(t)$: (1) $F(t) \geq 8\max(\gamma/\eta, 32\lambda/\eta^2)$, for which we show that the resulting approximation satisfies a multiplicative error of $(1 \pm 4\eta)$, and (2) $F(t) \leq 8\max(\gamma/\eta, 32\lambda/\eta^2)$, for which we show that the resulting approximation has an additive error of $32\max(\gamma/\eta, 32\lambda/\eta^2)$. We now separately consider the two cases.

**Case (1).** $F(t) \geq 8\max(\gamma/\eta, 32\lambda/\eta^2)$. Let $i^* \in [L]$ be the largest $i$ s.t. $\frac{F(t)}{2^{i^*}} \geq 4\max(\gamma/\eta, 32\lambda/\eta^2)$. Note that by Lemma D.7, $(1-\eta)\frac{F(t)}{2^{i^*}} \leq F_{i^*}(t) \leq (1+\eta)\frac{F(t)}{2^{i^*}}$. Therefore by the definition of $i^*$, $F_{i^*}(t) \leq 2\frac{F(t)}{2^{i^*}} \leq 2 \cdot 8\max(\gamma/\eta, 32\lambda/\eta^2) = 16\max(\gamma/\eta, 32\lambda/\eta^2)$. Since in this case the noisy count $\hat{s}_{i^*}$ at timestep $t$ would not exceed $\tau$ by Lemma D.3, so COUNTING-DICT$_{i^*}$ will not output TOO-HIGH in the stream $i^*$. Then $\mathcal{S}_{i^*}[t] = \hat{s}_{i^*}(t)$), and by Lemma D.8, BinaryMechanism-CD will be accurate and $\mathcal{S}_{i^*}[t] \geq F_{i^*}(t) - \gamma$. Since $F_{i^*}(t) \geq (1-\eta)\frac{F(t)}{2^{i^*}} \geq 2\max(\gamma/\eta, 32\lambda/\eta^2)$ by Lemma D.7 and using the fact that $\eta < 0.5$:

$$\mathcal{S}_{i^*}[t] \geq 2\max(\gamma/\eta, 32\lambda/\eta^2) - \gamma \geq \max(\gamma/\eta, 32\lambda/\eta^2). \qquad (3)$$

Above we showed that $F_{i^*}(t) \leq 16 \max(\gamma/\eta, 32\lambda/\eta^2)$. Then by Lemma D.3, the noisy count from $i^*$ will not exceed $\tau = 16 \max\{\gamma/\eta, 32\lambda/\eta^2\} + 2\sqrt{2} \frac{(\log T + 1)^{3/2} \sqrt{W \log(20T \lceil \log T \rceil/\beta)}}{\sqrt{\rho}}$, so COUNTING-DICT$_{i^*}$ will not output TOO-HIGH. The only concern now is that the output from $i^*$ may not be output if the noisy count is smaller than $\max(\gamma/\eta, 32\lambda/\eta^2)$ (see Line 22 of Algorithm 1), but by Inequality (3), this is impossible because $\mathcal{S}_{i^*}[t] \geq \max(\gamma/\eta, 32\lambda/\eta^2)$. Therefore, Algorithm 1 with COUNTING-DICT$_i$ as the subroutine will produce a non-zero output, i.e. it will output some $\mathcal{S}_{i'}[t] \cdot 2^{i'}$ for some $i' \geq i^*$ instead of 0.

We now proceed in two steps: first, we derive a lower bound on the true count $F_{i'}(t)$ in substream $i'$; then, we bound the ratio between Algorithm 1's output $\mathcal{S}_{i'}[t] \cdot 2^{i'}$ and the true count $F(t)$, using $F_{i'}(t)$ as an intermediate quantity.

We start with the lower bound on $F_{i'}(t)$:

$$
\begin{aligned}
F_{i'}(t) &\geq \mathcal{S}_{i'}[t] - \gamma && \text{by Lemma D.8} \\
&\geq (1 - \eta)\mathcal{S}_{i'}[t] && \text{because } \gamma \leq \eta\mathcal{S}_{i'}[t] \\
&\geq 16\lambda/\eta^2 && \text{because } 32\lambda \leq \eta^2\mathcal{S}_{i'}[t] \text{ and } 0 < \eta < 0.5
\end{aligned}
$$

Next we bound the ratio between $\mathcal{S}_{i'}[t] \cdot 2^{i'}$ and $F(t)$. According to Lemma D.7, $(1 - \eta)\frac{F(t)}{2^{i'}} \leq F_{i'}(t) \leq (1 + \eta)\frac{F(t)}{2^{i'}}$. Then,

$$
\begin{aligned}
& \mathcal{S}_{i'}[t] \leq F_{i'}(t) + \gamma && \text{by Lemma D.8} \\
\implies \quad & \mathcal{S}_{i'}[t] \leq \frac{F_{i'}(t)}{1 - \eta} && \text{because } \gamma \leq \eta\mathcal{S}_{i'}[t] \\
& \qquad\;\; \leq \frac{1 + \eta}{1 - \eta}\frac{F(t)}{2^{i'}} && \text{by Lemma D.7} \\
& \qquad\;\; \leq (1 + 4\eta)\frac{F(t)}{2^{i'}} && \text{because } 0 < \eta < 0.5
\end{aligned}
$$

To see the second inequality, we use the fact that $\gamma \leq \eta\mathcal{S}_{i'}[t]$, and plug this into $\mathcal{S}_{i'}[t] \leq F_{i'}(t) + \gamma$ to get $\mathcal{S}_{i'}[t] \leq F_{i'}(t) + \eta\mathcal{S}_{i'}[t]$. Rearranging gives that $\mathcal{S}_{i'}[t] \leq \frac{F_{i'}(t)}{1 - \eta}$. The third inequality is from Lemma D.7, which gives that $F_{i'}(t) \leq (1 + \eta)\frac{F(t)}{2^{i'}}$ whenever $F(t) \geq 2^{i'} \cdot \frac{4\lambda}{\eta^2}$. We now show that because we are in Case 1 where $F(t) \geq 8 \max(\gamma/\eta, 32\lambda/\eta^2)$, then it must always be the case that $F(t)/2^{i'} \geq \frac{4\lambda}{\eta^2}$. Assume towards a contradiction that $F(t)/2^{i'} < 4\lambda/\eta^2$. Then by Lemma D.7 $F_{i'}(t) \leq F(t)/2^{i'} + 4\lambda/\eta$. By the lower bound above $F_{i'}(t) \geq 16\lambda/\eta^2$. Combining these gives that,

$$
16\lambda/\eta^2 \leq F_{i'}(t) \leq F(t)/2^{i'} + 4\lambda/\eta < 4\lambda/\eta^2 + 4\lambda/\eta \leq 8\lambda/\eta^2,
$$

where the second to last step is from the assumption that $F(t)/2^{i'} < 4\lambda/\eta^2$ and the last step is because $\eta \in (0, 0.5)$. Clearly this is a contraction, so it must be that $F(t)/2^{i'} \geq \frac{4\lambda}{\eta^2}$.

By symmetric arguments,

$$
\mathcal{S}_{i'}[t] \geq F_{i'}(t) - \gamma \geq \frac{F_{i'}(t)}{1 + \eta} \geq \frac{1 - \eta}{1 + \eta}\frac{F(t)}{2^{i'}} \geq (1 - 4\eta)\frac{F(t)}{2^{i'}}.
$$

Thus in the case when $F(t) > 8 \max(\gamma/\eta, 32\lambda/\eta^2)$, Algorithm 1 produces a numerical output $\mathcal{S}_{i'}[t] \cdot 2^{i'}$ for some $i' > 0$, which will achieve a multiplicative error of $1 \pm 4\eta$ with respect to the true count $F(t)$.

**Case (2).** $F(t) \leq 8 \max(\gamma/\eta, 32\lambda/\eta^2)$, in which case again $F_i(t) \leq 16 \max(\gamma/\eta, 32\lambda/\eta^2)$ for all $i$, so the noisy count would not exceed $\tau$ by Lemma D.3, so COUNTING-DICT$_i$ will not output TOO-HIGH. Then, Algorithm 1 either outputs 0, which will result in additive error at most $8 \max(\gamma/\eta, 32\lambda/\eta^2))$, or it outputs $\mathcal{S}_{i'}[t] \cdot 2^{i'}$ for some $i'$, such that $\mathcal{S}_{i'}[t] \geq \max(\gamma/\eta, 32\lambda/\eta^2)$. In this latter case, by the same argument as in Case (1),

$$
F_{i'}(t) \geq \mathcal{S}_{i'}[t] - \gamma \geq (1 - \eta)\mathcal{S}_{i'}[t] \geq 16\lambda/\eta^2.
$$

Having established a lower bound for $F_{i'}(t)$, we now turn to derive an upper bound for the algorithm's output $\mathcal{S}_{i'}[t] \cdot 2^{i'}$. By Lemma D.7, we know that if $\frac{F(t)}{2^{i'}} \geq \frac{4\lambda}{\eta^2}$ then $F_{i'}(t) \leq 2\frac{F(t)}{2^{i'}}$, and otherwise, if $\frac{F(t)}{2^{i'}} < \frac{4\lambda}{\eta^2}$, then $16\lambda/\eta \leq 16\lambda/\eta^2 \leq$

$F_{i'}(t) \leq \frac{F(t)}{2^{i'}} + \frac{4\lambda}{\eta}$ which implies that $\frac{F(t)}{2^{i'}} \geq 12\lambda/\eta$. Thus in this case $F_{i'}(t) \leq \frac{F(t)}{2^{i'}} + 4\lambda/\eta \leq 2\frac{F(t)}{2^{i'}}$ as well, since $F(t)/2^{i'} \geq 12\lambda/\eta$. Hence under both conditions,

$$
\begin{aligned}
\mathcal{S}_{i'}[t] \cdot 2^{i'} &\leq (F_{i'}(t) + \gamma) \cdot 2^{i'} && \text{by Lemma D.8} \\
&\leq \frac{F_{i'}(t)}{1 - \eta} \cdot 2^{i'} && \text{because } \gamma \leq \eta \mathcal{S}_{i'}[t] \\
&\leq 2 \cdot 2 \cdot \frac{F(t)}{2^{i'}} \cdot 2^{i'} && \text{because } F_{i'}(t) \leq 2\frac{F(t)}{2^{i'}} \text{ and } 0 < \eta < 1/2 \\
&= 4F(t) \\
&\leq 32\max(\gamma/\eta, 32\lambda/\eta^2) && \text{by Case 2 condition}
\end{aligned}
$$

Therefore, in this case Algorithm 1 achieves an additive error of at most $32\max(\gamma/\eta, 32\lambda/\eta^2)$.

**When $ob$ is false:** This proof is very similar to the case where $ob$ is true. The main difference is that we need one extra lemma about the size of the blocklist. There is also an additional error caused by the blocklist, which can be treated as part of the error from the binary mechanism. The majority of the analysis stays the same; the only difference is that we will have a larger error in this case because of the blocklist.

We will use the three same key lemmas as in the case where $ob$=true, and an additional lemma bounding the size of the blocklist. These four lemmas will ensure that at a fixed timestep $t$, the desired good events will happen with high probability:

1. In all substreams $i \in [L]$, the correct number of distinct elements in the substream $i$ by hashing, denoted $F_i(t)$, is also a good estimator for the number of distinct elements in the entire stream at timestep $t$, denoted $F(t)$ (Lemma D.7). That is, for all $i \in [L]$, the following two conditions hold at the same time for any specific timestep $t$ with probability at least $1 - \beta/5$ for any $\eta \in (0, 0.5)$:

   (a) $\forall i \in [L]$ with $F(t) \geq 2^i \cdot \frac{4\lambda}{\eta^2}$, we have $(1 - \eta)\frac{F(t)}{2^i} \leq F_i(t) \leq (1 + \eta)\frac{F(t)}{2^i}$

   (b) $\forall i \in [L]$ with $F(t) < 2^i \cdot \frac{4\lambda}{\eta^2}$, we have $\frac{F(t)}{2^i} - \frac{4\lambda}{\eta} \leq F_i(t) \leq \frac{F(t)}{2^i} + \frac{4\lambda}{\eta}$.

2. BinaryMechanism-CD (Algorithm 5) is accurate (Lemma D.8). That is, for all $i \in [L]$, we have $|F_i(t) - \hat{s}_i(t)| \leq \gamma = \sqrt{\frac{4(W+1)(\log T+1)^3 \log(10(\log T+1)/\beta)}{\rho}}$ with probability at least $1 - \beta/5$.

3. For any stream $i$, if the correct number of distinct elements in the subtream $i$ is below a certain threshold then COUNTING-DICT$_i$ will not output TOO-HIGH (Lemma D.3). We plug $\beta/5L$ into Lemma D.3 to obtain that, if $F_i(t) \leq 16\max\{\gamma/\eta, 32\lambda/\eta^2\}$, then COUNTING-DICT$_i$ will not output TOO-HIGH, i.e. the noisy count $\hat{s}_i \leq \tau = 16\max\{\gamma/\eta, 32\lambda/\eta^2\} + 2\sqrt{2}\frac{(\log T+1)^{3/2}\sqrt{W\log(20T\lceil\log T\rceil/\beta)}}{\sqrt{\rho}}$.

4. The blocklist has bounded size (Lemma D.9). With probability at least $1 - \beta/5$, the size of the blocklist is bounded by $3T^{1/3}\log(T^{1/3}\lceil\log T\rceil/\beta)$.

Conditioned on these good events and plugging in $W = T^{2/3}$, by Lemmas D.8 and D.9, for each instance of BinaryMechanism-CD, the overall additive error stemming from the binary mechanism and blocklisting is $\gamma = \sqrt{\frac{4(T^{2/3}+1)(\log T+1)^3 \log(10(\log T+1)/\beta)}{\rho}} + 3T^{1/3}\log(T^{1/3}\lceil\log T\rceil/\beta)$.

To prove the desired accuracy claim, we will condition on all four high-probability events described above occurring at timestep $t$. Note that each of the four events occur with probability $1 - \beta/5$. Thus all four events will happen with probability at least $1 - \frac{4}{5}\beta \geq 1 - \beta$ by a union bound.

From here, we can again treat separately two cases based on $F(t)$, the number of distinct elements of the stream at time $t$: (1) $F(t) \geq 8\max(\gamma/\eta, 32\lambda/\eta^2)$ and (2) $F(t) \leq 8\max(\gamma/\eta, 32\lambda/\eta^2)$. As in the case where $ob$=true, we show in Case (1), the resulting approximation satisfies a multiplicative error of $(1 \pm 4\eta)$, and in Case (2), the resulting approximation has an additive error of $32\max(\gamma/\eta, 32\lambda/\eta^2)$. The analysis in both cases is identical to when $ob$=true as presented above, with the correspondingly larger value of $\gamma$, and so is not repeated here.

$\square$

## D.4. Proofs of Helper Lemmas for Theorem 4.6

Lemma D.7 bounds the number of elements in the substream after hashing. Lemma D.8 proves the accuracy of BinaryMechanism-CD algorithm. Lemma D.9 bounds the size of the blocklist when $ob$ is false. With the help of these lemmas, we can show the accuracy of COUNTING-DICT, as an intermediate step in the analysis.

**Lemma D.7** (Substream concentration bound). *Let $F(t)$ be the number of distinct elements of the stream at time $t$, and let $F_i(t)$ be the number of distinct elements of substream $\mathcal{S}_{i,g}$ (Definition D.1) for any $i \in [L]$. If $F(t) \geq 2^i \cdot \frac{4\lambda}{\eta^2}$, then $\Pr[|F_i(t) - \frac{F(t)}{2^i}| > \eta \cdot \frac{F(t)}{2^i}] \leq \frac{\beta}{5L}$. Otherwise, if $F(t) \leq 2^i \cdot \frac{4\lambda}{\eta^2}$, then $\Pr[|F_i(t) - \frac{F(t)}{2^i}| > \frac{4\lambda}{\eta}] \leq \frac{\beta}{5L}$. This implies that the following two conditions hold at the same time with probability at least $1 - \beta/5$:*

1. *$\forall i \in [L]$ with $F(t) \geq 2^i \cdot \frac{4\lambda}{\eta^2}$, we have $(1-\eta)\frac{F(t)}{2^i} \leq F_i(t) \leq (1+\eta)\frac{F(t)}{2^i}$*

2. *$\forall i \in [L]$ with $F(t) < 2^i \cdot \frac{4\lambda}{\eta^2}$, we have $\frac{F(t)}{2^i} - \frac{4\lambda}{\eta} \leq F_i(t) \leq \frac{F(t)}{2^i} + \frac{4\lambda}{\eta}$.*

*Proof.* We start with the case of $F(t) \geq 2^i \cdot \frac{4\lambda}{\eta^2}$. Applying Lemma A.5 to the $F_i(t)$ as a sum of $F(t)$ $\lambda$-wise independent $Bernoulli(2^{-i})$ random variables, and with $\mu = \frac{F(t)}{2^i}, A = \eta \cdot \frac{F(t)}{2^i}$ yields the following:

$$
\Pr\left[|F_i(t) - \frac{F(t)}{2^i}| > \eta \cdot \frac{F(t)}{2^i}\right] < 8\left(\frac{\lambda \cdot \frac{F(t)}{2^i} + \lambda^2}{\eta^2 \cdot (\frac{F(t)}{2^i})^2}\right)^{\lambda/2}
$$
$$
= 8\left(\frac{\lambda}{\eta^2 \frac{F(t)}{2^i}} + \frac{\lambda^2}{\eta^2(\frac{F(t)}{2^i})^2}\right)^{\lambda/2}
$$
$$
\leq 8\left(\frac{\lambda}{\eta^2(4\lambda/\eta^2)} + \frac{\lambda^2}{\eta^2(4\lambda/\eta^2)^2}\right)^{\lambda/2}
$$
$$
= 8\left(\frac{1}{4} + \frac{\eta^2}{16}\right)^{\lambda/2},
$$

where the third step is because of the case $F(t) \geq 2^i \cdot \frac{4\lambda}{\eta^2}$, and the fourth step simplifies terms.

We now wish to bound the right hand side by $\frac{\beta}{5L}$; solving this inequality for $\lambda$ yields the following bound:

$$
8\left(\frac{1}{4} + \frac{\eta^2}{16}\right)^{\lambda/2} \leq \frac{\beta}{5L}
$$
$$
\iff \left(\frac{1}{4} + \frac{\eta^2}{16}\right)^{\lambda/2} \leq \beta/40L
$$
$$
\iff \frac{\lambda}{2}\log(1/4 + \eta^2/16) \leq \log(\beta/40L)
$$
$$
\iff \lambda \geq \frac{2\log(\beta/40L)}{\log(1/4 + \eta^2/16)}
$$
$$
\iff \lambda \geq \frac{2\log(40L/\beta)}{\log(\frac{1}{1/4+\eta^2/16})}
$$

The last step comes from multiplying both the numerator and denominator by $-1$, and $-\log(x) = \log(1/x)$.

Since $\eta < 0.5$, then the denominator can be bounded by: $\log(\frac{1}{1/4+\eta^2/16}) > \log(64/17) > 1$. Since Algorithm 1 sets $\lambda = 2\log(40L/\beta)$, the above inequality will be satisfied.

Next we consider the second case, where $F(t) < 2^i \cdot \frac{4\lambda}{\eta^2}$. Applying Lemma A.5 again to the $F_i(t)$, now with $A = \frac{4\lambda}{\eta}$, yields

the following:

$$\Pr\left[|F_i(t) - \frac{F(t)}{2^i}| > \frac{4\lambda}{\eta}\right] < 8\left(\frac{\lambda \cdot \frac{F(t)}{2^i} + \lambda^2}{(\frac{4\lambda}{\eta})^2}\right)^{\lambda/2}$$

$$< 8\left(\frac{\lambda \cdot 4\lambda/\eta^2 + \lambda^2}{16\lambda^2/\eta^2}\right)^{\lambda/2}$$

$$= 8\left(\frac{1}{4} + \frac{\eta^2}{16}\right)^{\lambda/2},$$

where the second step is because of the case $F(t) < 2^i \cdot \frac{4\lambda}{\eta^2}$, and the third step simplifies terms.

We again wish to bound the right hand side by $\frac{\beta}{5L}$, and by the same steps as in the first case, the desired inequality holds if and only if $\lambda \geq \frac{2\log(40L/\beta)}{\log(\frac{1}{1/4+\eta^2/16})}$, and using the requirement that $\eta < 0.5$, Algorithm 1's choice of $\lambda = 2\log(40L/\beta)$ will satisfy the desired inequality. $\qquad\square$

**Lemma D.8** (Binary mechanism accuracy). *Fix a timestep $t \in [T]$, and recall that $\hat{s}_i(t)$ is the noisy count of $F_i(t)$ produced by BinaryMechanism-CD$_i$. Then $|F_i(t) - \hat{s}_i(t)| \leq \sqrt{\frac{4(W+1)(\log T+1)^3 \log(10(\log T+1)/\beta)}{\rho}}$ simultaneously for all $i \in [L]$ with probability $1 - \beta/5$.*

*Proof.* We will show that for a specific substream $i \in [L]$, the error $|F_i(t) - \hat{s}_i(t)|$ is bounded by the desired term with probability $1 - \beta/5L$, and the lemma will follow by a union bound over all $i \in [L]$.

Now consider the error of substream $i$ at timestep $t$. Algorithm 5 (BinaryMechanism-CD) adds a total of $\mathsf{Bin}_1(t)$ independent Gaussian noise terms, where $\mathsf{Bin}_1(t)$ is the number of ones in the binary representation of $t$. Hence, the error $|F_i(t) - \hat{s}_i(t)|$ is a sum of at most $\log T + 1$ independent Gaussian random variables each distributed as $\mathcal{N}(0, 2L(W+1)(\log T+1)/\rho)$, and so the error itself is also Gaussian with mean 0 and variance at most $\frac{2L(W+1)(\log T+1)^2}{\rho}$.

Applying a Chernoff bound (Lemma A.7) to this random variable, gives that for any $\gamma > 0$,

$$\Pr\left[|F_i(t) - \hat{s}_i(t)| > \gamma\right] \leq 2\exp\left(\frac{-\rho\gamma^2}{4L(W+1)(\log T+1)^2}\right).$$

We wish to bound this probability by $\beta/5L$. Solving this inequality for $\gamma$ yields $\gamma \geq \sqrt{\frac{4L(W+1)(\log T+1)^2 \log(10L/\beta)}{\rho}}$. Since $L = \lceil\log T\rceil \leq \log T + 1$, then choosing $\gamma = \sqrt{\frac{4(W+1)(\log T+1)^3 \log(10(\log T+1)/\beta)}{\rho}}$ will satisfy the inequality. $\qquad\square$

**Lemma D.9** (Bounded blocklist size). *Suppose $ob = false$. Fix a timestep $t \in [T]$. With probability at least $1 - \beta/5$, the size of the blocklist $\mathcal{B}$ is bounded by $3T^{1/3}\log(T^{1/3}L/\beta)$, where $L = \lceil\log T\rceil$.*

*Proof.* The size of the blocklist is non-decreasing since elements are never removed, so it is sufficient to upper bound the final size of the blocklist after $T$ timesteps.

Define the random variable $Y_j$ to be 1 if the $j$-th arrival in the stream is blocklisted and 0 if otherwise. Then $Y = \sum_{i=j}^{T} Y_j$ is an upper bound on the size of the blocklist (because the arrivals may be updates of the same element). The sampling rate for blocklisting is $p = \frac{\log(T^{1/3}L/\beta)}{T^{2/3}}$, so $Y_j \sim Bern(\frac{\log(T^{1/3}L/\beta)}{T^{2/3}})$ and $\mathbb{E}[Y] = \frac{T\log(T^{1/3}L/\beta)}{T^{2/3}} = T^{1/3}\log(T^{1/3}L/\beta)$. Applying a multiplicative Chernoff bound (Theorem A.6) with $\eta = 2$ yields:

$$\Pr\left[Y > 3T^{1/3}\log(T^{1/3}L/\beta)\right] \leq \exp\left(-T^{1/3}\log(T^{1/3}L/\beta)\right)$$

We would like to bound this probability by $\beta/5$, which occurs when $T^{1/3}\log(\frac{T^{1/3}L}{\beta}) \geq \log(\frac{5}{\beta})$. This inequality will clearly hold if $T^{1/3}L = T^{1/3}\lceil\log T\rceil \geq 5$, which is true for $T \geq 8$. $\qquad\square$

### D.5. Space Complexity

We restate Theorem 4.7 below for convenience.

**Theorem 4.7.** *With probability at least* $1 - \beta$, *assuming the universe size* $|\mathcal{U}| = poly(T)$: *If ob is true, the space complexity of Algorithm 1 is* $O(\sqrt{W} \cdot polylog(T/\beta)) \cdot poly(\frac{1}{\rho\eta})$. *If ob is false, the space complexity of Algorithm 1 is* $O(T^{1/3} \cdot polylog(T/\beta)) \cdot poly(\frac{1}{\rho\eta})$.

*Proof of Theorem 4.7.* We condition on all the high-probability events used in the proof of Theorem 4.5; these events occur with probability $1 - \beta$ as shown in the proof of Theorem 4.5. The space usage comes from (1) the KSET data structure in COUNTING-KSET, (2) BinaryMechanism-CD, and (3) the blocklist (when *ob* is false).

- From Lemma B.1, the $L$ instantiations of COUNTING-KSET together have space complexity $L \cdot O(k(\log T + \log|\mathcal{U}|)\log(k/\beta)) = O(\log T \cdot k \cdot (\log T + \log|\mathcal{U}|)\log(k/\beta))$ where $\lambda = 2\log(40\lceil\log(T)\rceil/\beta)$ and $k = 16\max\{\gamma/\eta, 32\lambda/\eta^2\} + 4\sqrt{2}\frac{(\log T+1)^{3/2}\sqrt{W\log(20T\lceil\log T\rceil/\beta)}}{\sqrt{\rho}}$ (here $W = T^{2/3}$ when *ob* is false). We make the standard assumption that the universe size $|\mathcal{U}|$ is $poly(T)$, so we can represent an item in $poly(T)$ bits. This means that the space complexity simplifies to $O(k \cdot poly\log(T) \cdot \log(k/\beta))$.

- As shown in (Chan et al., 2011), one instance of BinaryMechanism-CD uses $O(\log(T))$ space, and therefore the $L$ copies of the BinaryMechanism-CD use space $O(L \cdot \log(T)) = O(\log^2(T))$.

- The blocklist when *ob* is false has size $O(T^{1/3}\log(T/\beta))$ when conditioned on the high-probability events.

The dominating term is the one from the KSET data structures, which is $O(k \cdot poly\log(T) \cdot \log(k/\beta))$ for $k = 16\max\{\gamma/\eta, 32\lambda/\eta^2\} + 4\sqrt{2}\frac{(\log T+1)^{3/2}\sqrt{W\log(20T\lceil\log T\rceil/\beta)}}{\sqrt{\rho}}$.

If *ob* is true, $\gamma = \sqrt{\frac{4(W+1)(\log T+1)^3\log(10(\log T+1)/\beta)}{\rho}}$ in which case the space complexity simplifies to $O(\sqrt{W} \cdot poly\log(T/\beta)) \cdot poly(\frac{1}{\rho\eta})$; otherwise, when *ob* is false, $\gamma = \sqrt{\frac{4(T^{2/3}+1)(\log T+1)^3\log(10(\log T+1)/\beta)}{\rho}} + 3T^{1/3}\log(5T^{1/3}\lceil\log T\rceil/\beta)$, in which case the space complexity simplifies to $O(T^{1/3} \cdot poly\log(T/\beta)) \cdot poly(\frac{1}{\rho\eta})$. $\qquad\square$

## E. Blocklisting Problem: Space Upper and Lower Bounds

In this section, we formally define the problems of blocklisting items with high flippancy (and high occurrency) and prove a space lower bound for both problems. Our lower bound is information theoretic and applies to any algorithm for blocklisting (flippancy or occurrency), including exponential time algorithms and non-private algorithms. Then, we show that the problem of blocklisting occurrency has an almost-matching space upper bound that is tight up to log factors, given by Algorithm 1.

Recall that the flippancy of an item is defined in (Jain et al., 2023) as the number of timesteps where the item switches between being present to absent or vice versa, while occurrency is defined by the number of timesteps where an item appears in the stream (with any sign).

Informally, we define the $\mathbf{blocklist_{flip}(W)}$ (resp., $\mathbf{blocklist_{occ}(W)}$) problem to be the problem of identifying, for each timestep of a turnstile stream, whether the current element of the stream has flipped $< W$ times (resp., occurred $< W$ times) before this timestep. More formally, let $\mathcal{U}$ be a universe of items and let $x = (x_1, \ldots, x_T)$, be a turnstile stream where for each time $t \in [T]$, the stream element $x_t$ is either an insertion $(+u)$ or deletion $(-u)$ for some $u \in \mathcal{U}$, or $x_t = \perp$ indicating an empty update.

**Definition E.1** (The $\mathbf{blocklist_{flip}(W)}$ (resp., $\mathbf{blocklist_{occ}(W)}$) problem)**.** For the turnstile stream $x = (x_1, \ldots, x_T)$, define the *ground truth* for $x$ as the binary stream of outputs $o^*(x) = (o_1^*, \ldots, o_T^*)$ where for each $t \in [T]$, $o_t^* = 0$ if $x_t$ has flippancy $< W$ (resp., occurrency $< W$) in the prefix stream $(x_1, \ldots, x_{t-1})$ or if $x_t = \perp$, and $o_t^* = 1$ otherwise. Let $o(x) = (o_1, \ldots, o_T)$ be the output provided by an algorithm on the stream $x$. The algorithm has a *false negative* at time $t$ if $o_t = 0$ and $o_t^* = 1$, and has a *false positive* when $o_t = 1$ and $o_t^* = 0$.

Notice that algorithms using flippancy blocklisting such as (Jain et al., 2023), or occurrency blocklisting like our algorithm are required to have no false negatives with high probability. This is because the max flippancy (and occurrency) bound is

used to upper bound the sensitivity of the binary tree mechanism and thus needs to hold with probability at least $(1 - \delta)$ to achieve $(\varepsilon, \delta)$-DP.

While false negatives affect the privacy of the algorithm, false positives must also be bounded to ensure accuracy. For this reason, we ask if it is possible to design *low-space* blocklisting algorithms (for flippancy or occurrency) that guarantee (with high probability) no false negatives, while bounding the number of false positives. In Section E.1, we prove a lower bound on the space of any algorithm that bounds flippancy or occurrency. Then in Section E.2, we give a near-matching upper bound by showing that Algorithm 1 solves the low-space occurrency blocklisting problem using space that matches the lower bound up to log-factors.

### E.1. Lower bound

We first show that any algorithm (including non-private and exponential-time algorithms) for $\mathbf{blocklist_{flip}(W)}$ and $\mathbf{blocklist_{occ}(W)}$, with no false negatives and with bounded false positives must have a space that depends on the number of false positives allowed. This space lower bound also extends to any algorithm for count distinct estimation that uses blocklisting methods to control flippancy or occurrency, such as (Jain et al., 2023) or our work.

**Theorem E.2.** *For any even integer $W > 0$, and any integer $r > 0$, let $\mathcal{A}_{flip}$ (resp., $\mathcal{A}_{occ}$) be an algorithm for $\mathbf{blocklist_{flip}(W)}$ (resp., $\mathbf{blocklist_{occ}(W)}$) such that, given an arbitrary stream $x$ of length $T$, with probability at least $1 - \beta$, has no false negatives and has at most $r$ false positives. Then algorithm $\mathcal{A}_{flip}$ and $\mathcal{A}_{occ}$ use space at least:*

$$(1 - \beta) \cdot \left( \log(1 - \beta) + \frac{T}{2W} \log \frac{WT}{T + 2Wr} \right).$$

We provide a proof sketch here, and a full proof is deferred to Appendix F.1. Our lower bound proceeds by describing a random process defining a stream distribution that is hard for the problems of $\mathbf{blocklist_{flip}(W)}$ and $\mathbf{blocklist_{occ}(W)}$. For the element universe $\mathcal{U} = [T/2]$, the main idea is to define a distribution of problem instances as follows. First, generate $X \subseteq \mathcal{U}$ as a uniformly random set of size $T/2W$. Then define a stream $x(X)$ where for the first $T/2$ timesteps, each element $u \in X$ appears in $W$ updates, alternating $W/2$ times between one insertion and one deletion of $u$. This results in $W$ flippancy and $W$ occurrency for all elements of $X$ at the end of the first half of the stream. In the second $T/2$ timesteps of the stream, all elements in $\mathcal{U}$ are inserted once. Thus the correct output for this stream for both the $\mathbf{blocklist_{flip}(W)}$ and $\mathbf{blocklist_{occ}(W)}$ problems is to always output 0 in the first half, and to output 1 in the second half only for elements in $X$.

We then use an information theory argument to show a space lower bound for any algorithm $\mathcal{A}_{flip}$ that satisfies the conditions of no false negatives and at most $r$ false positives, with probability at least $1 - \beta$ over this distribution of problem instances.

### E.2. Upper bound for occurrency blocklisting

We now present an upper bound (Corollary E.3) based on Algorithm 1 for the space required to solve the occurrency blocklisting problem with $r$ false positives.

**Corollary E.3.** *With probability $1 - 2\beta$ and when $|\mathcal{U}| = poly(T)$, Algorithm 1 with ob=false reports no false negatives and $r = 2T^{1/3} \log(T^{1/3} \lceil \log(T) \rceil / \beta)$ false positives for the problem $\mathbf{blocklist_{occ}(W)}$ for $W = T^{2/3}$, while using space $O(T^{1/3} \cdot polylog(T/\beta)) \cdot poly(\frac{1}{\rho\eta}))$.*

Note that plugging $r = 2T^{1/3} \log(T^{1/3} \lceil \log(T) \rceil / \beta)$ and $W = T^{2/3}$ into the lower bound of Theorem E.2 gives an near-matching space lower bound that is tight up to log factors.

The proofs for no false negatives and the space complexity follow from Lemma D.5 in Appendix D.2, which gives a high probability bound on the maximum occurrency of the stream after blocklisting, and from Theorem 4.7, which bounds the space used by Algorithm 1. The proof for the bounded number of false positives follows from a concentration bound the probability of blocklisting an element too early. The full proof is deferred to Appendix F.2.

## F. Omitted Proofs from Appendix E

### F.1. Proof of Theorem E.2

**Theorem E.2.** *For any even integer $W > 0$, and any integer $r > 0$, let $\mathcal{A}_{flip}$ (resp., $\mathcal{A}_{occ}$) be an algorithm for $\mathbf{blocklist_{flip}(W)}$ (resp., $\mathbf{blocklist_{occ}(W)}$) such that, given an arbitrary stream $x$ of length $T$, with probability at*

*least* $1 - \beta$, *has no false negatives and has at most* $r$ *false positives. Then algorithm* $\mathcal{A}_{flip}$ *and* $\mathcal{A}_{occ}$ *use space at least:*

$$(1 - \beta) \cdot \left( \log(1 - \beta) + \frac{T}{2W} \log \frac{WT}{T + 2Wr} \right).$$

*Proof.* We now define a random process defining a stream distribution that is simultaneously hard for the the problem of **blocklist**$_\mathbf{flip}$(**W**) and for **blocklist**$_\mathbf{occ}$(**W**). For notational convenience, set $n = T/2$, and $m = n/W = T/(2W)$. For simplicity, we assume $T, W$ such that $n, m$ are integers.

Let the element universe be $\mathcal{U} = [n]$, over which we assume there is a total order. We define a distribution of problem instances with the following process:

- A uniformly random set $X \subseteq \mathcal{U}$ of size $m$ is sampled.

- Generate the first $n = T/2$ timesteps ensuring that all elements in $X$ have $W$ flippancy and $W$ occurrency.

- Then generate the second $n = T/2$ timesteps ensuring that all elements in $\mathcal{U}$ are inserted once.

Define a stream $x(X)$ to be a deterministic function of $X$ as follows. For the first half, all updates related to each element in $X$ appear consecutively and in the total order. Each element $u \in X$ has exactly $W$ updates, alternating $W/2$ times between one insertion and one deletion of $u$; this results in both $W$ flippancy and $W$ occurrency for all $u \in X$ at the end of the first half of the stream. Then, the second half of the stream has one insertion of each element in $\mathcal{U}$, appearing according to the total order.

It is easy to see that the correct output for this stream in both the **blocklist**$_\mathbf{flip}$(**W**) and **blocklist**$_\mathbf{occ}$(**W**) problems is to always output 0 in the first half, and to output 1 in the second half only for elements in $X$: $o^*(x(X))_t = 0$ for $t \leq n$ and $o^*(x(X))_t = 1$ iff $x_t \in X$ and $t \geq n + 1$.

We want to show a space lower bound for an algorithm $\mathcal{A}_{flip}$ (resp., $\mathcal{A}_{occ}$) that will satisfy the conditions of no false negatives, and at most $r$ false positives, with probability at least $1 - \beta$ over this distribution, for the **blocklist**$_\mathbf{flip}$(**W**) (resp., **blocklist**$_\mathbf{occ}$(**W**)) problem. The proof proceeds identically for both problems, so we prove it only for $\mathcal{A}_{flip}$.

Without loss of generality, we assume $\mathcal{A}_{flip}$ is deterministic, since for any randomized algorithm, there will exist a random seed to achieve no worse guarantee on the considered input distribution.

We prove this space lower bound via an information theory argument (see Section A.4 for information theory basics) by considering three random variables in addition to $X$:

- $S$: A random variable representing the memory state of algorithm $\mathcal{A}_{flip}$ after observing the first $n = T/2$ timesteps.

- $Y$: The set of elements with an output 1 in the second half of the stream by algorithm $\mathcal{A}_{flip}$.

- $P$: An indicator variable that is 1 if $\mathcal{A}_{flip}$ has no false negatives and at most $r$ false positives, and 0 otherwise. By definition we know $\Pr[P = 1] \geq 1 - \beta$.

Since the algorithm $\mathcal{A}_{flip}$ is deterministic, then $S, Y, P$ are all deterministic functions of $X$, since the stream itself is a deterministic function of $X$. Since algorithm $\mathcal{A}_{flip}$ does not learn new information in the second half of the stream – since the second half of the stream is the same for all streams – then $Y$ is a deterministic function of $S$.

To start, the maximum size of $S$ can be lower bounded by its entropy $H(S)$. Then $H(S)$ can be further lower bounded by the mutual information between the input secret $X$ and the random variable $Y$, which is part of algorithm $\mathcal{A}_{flip}$'s output, conditioned on $P$. This can be seen using inequalities from Fact A.12:

$$|S| \geq H(S) \geq H(S|P) \geq I(X; S|P) \geq I(X; Y|P).$$

Next, we apply the definitions of condition mutual information and mutual information:

$$\begin{aligned} I(X; Y|P) &= \Pr[P = 1] \cdot I(X; Y|P = 1) + \Pr[P = 0] \cdot I(X; Y|P = 0) \\ &\geq \Pr[P = 1] \cdot I(X; Y|P = 1) \\ &= \Pr[P = 1] \cdot (H(X|P = 1) - H(X|P = 1, Y)) . \end{aligned}$$

For $H(X|P = 1)$, since $\mathcal{A}_{flip}$ is deterministic and $\Pr[P = 1] \geq 1 - \beta$, we know that $X|P = 1$ is distributed uniformly across at least $(1 - \beta) \cdot \binom{n}{m}$ sets of universe elements[7] , and therefore,

$$H(X|P = 1) \geq \log\left((1 - \beta) \cdot \binom{n}{m}\right).$$

Next we analyze $H(X|P = 1, Y)$. Conditioned on $P = 1$, by the no false negatives requirement, we know $X \subseteq Y$, and by at most $r$ false positives, we know $|Y| \leq m + r$. Therefore, for any $y$ such that $\Pr[Y = y|P = 1] > 0$, it must be that $|y| \leq m + r$ and that conditioned on $P = 1$ and $Y = y$, $X$ is a subset of $y$ of size $m$. Therefore $X$ can take at most $\binom{m+r}{m}$ different values. By Fact A.12,

$$H(X|P = 1, Y = y) \leq \log\binom{m + r}{m}$$

and

$$H(X|P = 1, Y) = \sum_{y \subseteq [n]} \Pr[Y = y|P = 1] \cdot H(X|P = 1, Y = y) \leq \log\binom{m + r}{m}.$$

Putting everything together:

$$
\begin{aligned}
|S| \geq H(S) \geq I(X; Y|P) &\geq \Pr[P = 1] \cdot (H(X|P = 1) - H(X|P = 1, Y)) \\
&\geq (1 - \beta) \cdot \left(\log\left((1 - \beta) \cdot \binom{n}{m}\right) - \log\binom{m + r}{m}\right) \\
&= (1 - \beta) \cdot \left(\log(1 - \beta) + \log\frac{n \times (n - 1) \times \cdots \times (n - m + 1)}{(m + r) \times (m + r - 1) \times \cdots \times (r + 1)}\right) \\
&\geq (1 - \beta) \cdot \left(\log(1 - \beta) + m\log\frac{n}{m + r}\right) \\
&= (1 - \beta) \cdot \left(\log(1 - \beta) + \frac{T}{2W}\log\frac{WT}{T + 2Wr}\right).
\end{aligned}
$$

$\square$

## F.2. Proof of Corollary E.3

**Corollary E.3.** *With probability $1 - 2\beta$ and when $|\mathcal{U}| = poly(T)$, Algorithm 1 with ob=false reports no false negatives and $r = 2T^{1/3}\log(T^{1/3}\lceil\log(T)\rceil/\beta)$ false positives for the problem $\text{blocklist}_{\text{occ}}(\mathbf{W})$ for $W = T^{2/3}$, while using space $O(T^{1/3} \cdot polylog(T/\beta)) \cdot poly(\frac{1}{\rho\eta}))$.*

*Proof.* For the space complexity of Algorithm 1, Theorem 4.7 says that when $|\mathcal{U}| = poly(T)$ and *ob*=false, then with probability at least $1 - \beta$, Algorithm 1 uses space $O(T^{1/3} \cdot polylog(T/\beta)) \cdot poly(\frac{1}{\rho\eta})$.

For the claim that Algorithm 1 has no false negatives with high probability, Lemma D.5 in Appendix D.2 shows that when $ob = false$, with probability at least $1 - \beta/L$, the maximum occurrency of the stream produced from the blocklisting is at most $T^{2/3}$. Since Algorithm 1 uses $W = T^{2/3}$, then with probability at least $1 - \beta/L$, Algorithm 1 will have no false negatives, since it will never allow elements with occurrency larger than $W = T^{2/3}$ to persist without being blocklisted. Plugging in the value of $L = \lceil\log(T)\rceil$, note that $\beta/\lceil\log(T)\rceil \leq \beta/2$, therefore with probability $1 - \beta/2$, Algorithm 1 will have no false negatives.

To see the false positive bound, let $X_i$ be an indicator random variable where $X_i = 1$ if the element at timestep $i$ is a false positive in Algorithm 1, and let $X = \sum_{i=1}^{T} X_i$. Recall that Algorithm 1 samples elements for the blocklist with probability $p = \frac{\log(T^{1/3}L/\beta)}{T^{2/3}}$ at each occurrence. Note that $\Pr[X_i = 1] = \Pr[(x_i \in \mathcal{B}) \cap (x_i < W)] \leq \Pr[(x_i \in \mathcal{B})] = p$, so then

---

[7]To see this, first observe that $P(X) = 1$ partitions the input space according to the event that $\mathcal{A}_{flip}$ has no false negatives and at most $r$ false positives. Let $R$ denote the size of the support for $X|P = 1$. Then, since $X$ is uniformly chosen, $\Pr[P(X) = 1] \geq 1 - \beta$ means that $\frac{R}{\binom{n}{m}} \geq 1 - \beta$, and therefore $R \geq (1 - \beta) \cdot \binom{n}{m}$

$\mathbb{E}[X] = \sum_{i=1}^{T} \Pr[X_i = 1] \leq T \cdot p = T^{1/3} \log(T^{1/3} L/\beta)$. Applying a multiplicative Chernoff bound (Theorem A.6) with $\eta = 1$ yields:

$$\Pr[X \geq 2T^{1/3}\log(T^{1/3}L/\beta)] \leq \exp(-(T^{1/3}/3)\log(T^{1/3}L/\beta)) = \left(\frac{\beta}{T^{1/3}L}\right)^{T^{1/3}/3} \leq \beta/2.$$

To see the last inequality, we first plug in $L = \lceil\log(T)\rceil$, so we wish to show $\left(\frac{\beta}{T^{1/3}\lceil\log(T)\rceil}\right)^{T^{1/3}/3} \leq \beta/2$. We observe numerically that this holds for $T \geq 8$.

Taking a union bound over the failure probability from the space complexity, false positive, and false negative bounds, then with probability $1 - 2\beta$, Algorithm 1 solves the **blocklist**$_{occ}(W)$ for the desired $r$ and space conditions.

$\square$

