# OpenReview forum: "Differentially Private Space-Efficient Algorithms for Counting Distinct Elements in the Turnstile Model"
_ICML.cc/2025/Conference — ICML 2025 poster_

### Official Review · Reviewer_M6qz · 2025-03-09

**Overall Recommendation:** 3

**Summary:**

This paper studies the classical problem of counting distinct elements in a streaming (bounded space) setting, under differential privacy constraints in the continual release model (where the requirement is to provide an output at all times along the stream, while retaining differential privacy as usual). The main result is a $(1+\eta)$-approximation algorithm with $O(T^{1/3})$ space and additive error. It is known that a polynomial additive error is needed in this setting, from work of Jain, Kalemaj, Raskhodnikova, Sivakumar, and Smith in NeurIPS'23 (henceforth JKRSS) who showed an $\Omega(T^{1/4})$ additive error lower bound among other results. Notably, the JKRSS paper did not consider sublinear-space algorithms, and presented the question of obtaining such algorithms as an open problem, which the current paper resolves.

The main conceptual contribution of the current paper is a notion called the occurrency. The occurrency of a single element from the universe w.r.t. the stream is the number of times this element appears in the stream (both insertions and deletions count), and the maximum occurency in the stream is simply the maximum among all elements. Note that high occurrency items are simply heavy hitters (in an insertion only sense, where the number of occurrences always goes up, regardless of the sign).

To achieve the above bound, the authors combine an algorithm suitable for low-occurrency elements with a separate algorithm dealing with the heavy hitter elements that appear a lot of times (high occurrency items). One algorithm deals with all elements up to occurrency $W$, where $W$ can be an artificial upper bound (or simply the maximum occurrency), and achieves roughly $\sqrt{W}$ space and $\sqrt{W}$ additive error. The other algorithm deals with heavy hitters and has space bound $O(T/W)$. Setting $W = T^{2/3}$ achieves the aforementioned $O(T^{1/3})$ result. (Here I omitted lower order terms.)

The analogue notion used in JKRSS is that of flippancy. The flippancy of an item is the maximum number of items it flipped from 1 or -1 to 0 or vice versa. In their work they obtained an algorithm whose additive error is of order $\sqrt{F}$, where $F$ is the flippancy. Because the flippancy is bounded by the occurency and can be much smaller, the additive error of the JKRSS approach can be much better than the current paper. However, the space complexity of their approach is not sublinear, so the main saving here is in terms of space.

**Claims And Evidence:**

Yes, this is a theoretical paper and the claims are proved.

**Essential References Not Discussed:**

Occurency is very closely related to the notion of a a heavy hitter. An element has high occurency if and only if it is a heavy hitter, when we consider the turnstile stream an insertion only one. You should connect this notion to the heavy hitters literature and provide ample context.

Your result is very reminiscent of several dense-sparse tradeoffs works in robust streaming (that is, streaming against an adaptive adversary while providing continual output). It is well known by now that differential privacy is a central tool in robust streaming [1]. The DP-based approach of [1] achieves, for a global notion called flip number [2] (denote it by $\lambda$) (which differs somewhat from flippancy, though there are some similarities) roughly $\sqrt{\lambda)$ space and $\sqrt{\lambda}$ additive error (which is hidden in [1] in the computation of a DP median).

The dense-sparse robust streaming work of [3] achieves $T^{1/3}$ space complexity by combining the DP-approach with sparse recovery (whereas your work combines a similar variant of the former with a heavy hitters approach).
The work of [4] gets a dense-sparse tradeoff by using heavy hitters. I strongly recommend you to read [3] and [4] and see how your work compares to them. Your use of KSET seems to be novel.

[1] A. Hassidim, H. Kaplan, Y. Mansour, Y. Matias, U. Stemmer, "Adversarially robust streaming algorithms via differential privacy", JACM 2022. Preliminary version in NeurIPS 2020.
[2] O. Ben-Eliezer, R. Jayaram, D. Woodruff, E. Yogev, "A frame for adversarially robust streaming algorithms", JACM 2022, preliminary version in PODS 2020.
[3] O. Ben-Eliezer, T. Eden, K. Onak, "Adversarially robust streaming via dense--sparse trade-offs", SOSA 2022.
[4] D. Woodruff, S. Zhou, "Adversarially robust dense-sparse tradeoffs via heavy-hitters", NeurIPS 2024.

**Experimental Designs Or Analyses:**

None, this is a theoretical paper.

**Methods And Evaluation Criteria:**

NA

**Other Comments Or Suggestions:**

See "essential references not discussed".

**Other Strengths And Weaknesses:**

This paper is very well written -- I enjoyed reading it.

**Questions For Authors:**

I do not have questions for the authors; however I do think it is important that they address my comments in the "essential references not discussed" section.

I am not opposed to increasing your score, provided that you will be able to convince me that your approach is meaningfully different from what is done in the robust streaming literature (see references above).

**Relation To Broader Scientific Literature:**

The paper improves upon the Jain et al. paper in terms of space complexity. The results are quite similar to the dense-sparse tradeoffs literature in adversarially robust streaming, see detailed discussion in the next bullet.

**Theoretical Claims:**

I did not read the proofs, but I have a rough understanding of how to prove these results and am pretty sure they should be correct.

---

> ### Author Rebuttal · Authors · 2025-03-31
>
> We thank the reviewer for the thoughtful comments and valuable feedback.
>
> **Connections to heavy hitters:**
> We agree that there is a conceptual connection to the heavy hitters problem, as high-frequency elements can be viewed as heavy hitters, especially in the insertion-only setting. However, in the turnstile setting, an element may have high occurrency without being a heavy hitter in the frequency estimation sense: for example, if the element is very frequently added to and removed from the stream. As a result, the goal of our work is different from the heavy hitters problem: we are not trying to identify these high-occurrency elements, but simply to filter them out as items arrive. For this purpose, a simple probabilistic blocklisting approach suffices, and there is no need to apply a heavy hitters algorithm. Moreover, as shown by our lower bound, our blocklisting technique is space-optimal (up to log factors), so the potential improvements from using heavy-hitter algorithms as a form of blocklisting (e.g., use HH algorithms to identify high-occurency algorithms, and treating these as the blocklisted elements)  is limited. We will add a discussion to the final version of the paper on the connection of occurrency to the heavy hitters problem, as well as the differences that arise in the turnstile model specifically.
>
> **Differences from robust streaming:**
> The problem studied in this work and our approach to solving it are in fact different from the robust streaming literature. Here we highlight these differences in the context of lp-moment estimation. The goal of the robust streaming literature is to design algorithms that can give meaningful utility and (low) space guarantees for lp-moment estimation in the presence of an adaptive adversary, which can generate inputs to the stream based on previous timesteps and feed this to the algorithm in the next timestep. In this model, the algorithm can output an approximation at every timestep, but the **output does not need to preserve privacy** as in our DP algorithms. In contrast, in the continual release model, the algorithm is required to output a solution at every timestep and these outputs must also preserve DP. As a result, the two problem settings are not comparable and the solution for one problem cannot in general be used to solve the other problem (especially if efficient and accurate solutions are desired).
>
> The DP techniques introduced in [1] and subsequently used in works like [3,4] in the robust streaming literature are different from ours and are intended for a different purpose. Thus it is not possible to directly use prior work in the area to solve our problem (or vice versa). Importantly, the main DP technique introduced by [1] is used to preserve the privacy of the internal **randomness of the algorithm**,  but not to preserve the privacy of the algorithm’s **input**. As such, these algorithms do not protect the privacy of the user data in input: the adversary can infer information about the user’s data in the input, but is not able to learn about the random bits used (thus making adaptive attacks to the utility of the algorithm harder).
>
> We emphasize that the solutions of [3,4] do not preserve DP at every timestep. For example, [3] devises a solution for lp-moment frequency estimation by separating the cases of when the underlying frequency vector is sparse vs dense. For the sparse case, their algorithm stores the input explicitly using sparse recovery whereas for the dense case they use the DP technique from [1]. Thus for the sparse case, their output is non-private as it is directly computed from the sparse recovery structure.
>
> In contrast, because our goal is to preserve the privacy of the input stream at every timestep, our DP techniques need to differ from the ones used in the robust streaming literature. While we also use sparse recovery tools such as KSET, we must ensure that the output of the KSET does not leak privacy by analyzing its consistency properties (towards sensitivity analysis), and use DP building blocks adapted to our problem such as BM-CountDistinct (Algorithm 5).
>
> Regarding flippancy vs flip number, we emphasize that the concept of flippancy was introduced by [Jain et al] and we show in our work that it may not be a suitable measure to design low space algorithms in the DP turnstile model due to its stateful nature (L91-108), which is the main motivation for introducing occurrency instead. In general, the flip number is a property intrinsic to the problem being computed, while the flippancy (and occurrency) is a property intrinsic to the input stream (regardless of the problem). Thus these two measures are somewhat incomparable.

---

> > ### Comment · Reviewer_M6qz · 2025-04-02
> >
> > Thank you for the response. I will raise my score by one point, as it is a valuable contribution, but I do ask you to more explicitly mention the broader connections (in particular to heavy hitters, up to you re: robust streaming) discussed in the review.

---

> > > ### Author Response · Authors · 2025-04-03
> > >
> > > Thank you for your thoughtful feedback and for responding to our rebuttal. We appreciate your suggestions and will revise the paper to more explicitly highlight the broader connections to the heavy hitters and robust streaming literature. Thanks again!

---

### Official Review · Reviewer_F6kk · 2025-03-10

**Overall Recommendation:** 4

**Summary:**

The paper considers the problem of continually releasing private estimates of the number of distinct elements in a turnstile stream (with insertion and deletion of elements). It presents the first such algorithm that uses sublinear space while maintaining error similar to that of previous private algorithms (that use linear space).

## Update after rebuttal
The rebuttal confirms my assessment

**Claims And Evidence:**

The paper is theoretical. All claims are backed by proofs.

**Essential References Not Discussed:**

None that I am aware of.

**Experimental Designs Or Analyses:**

N/A

**Methods And Evaluation Criteria:**

There is no empirical evaluation.

**Other Comments Or Suggestions:**

-

**Other Strengths And Weaknesses:**

It appears that the paper could be simplified somewhat by substituting some of the building blocks. For example, the Misra-Gries algorithm (cited in the paper) could be used to keep track (separately) of the insertions and deletions in the stream, guaranteeing that all elements with occupancy (or flippancy) at least W are detected. This seems simpler than the current Theorem 1.3 and avoids the error probability.

In a similar vein, the KSET structure is not a modern solution. Invertible Bloom Lookup Tables, IBLTs (in various forms) give the functionality you need with good space usage, speed, and low error probability.

I have a few technical questions/concerns (questions below) but given the expected answers I do not think they are serious.

**Questions For Authors:**

- In Algorithm 1, line 22 you always output a multiple of a power of 2. I wonder how this is compatible with relative error arbitrarily close to 1?
- Do you assume that the hash functions used are fully random? If the space for the hash functions is not counted this requires some discussion.
- Would implementing the method with, say, universal hashing potentially impact privacy, utility, or both?

**Relation To Broader Scientific Literature:**

The paper is a timely follow-up to the recent work on tracking distinct elements under continual observation (NeurIPS '23), making progress on one of the main open problems from that work. This seems like a fundamental problem with multiple potential applications, but most past work has focused on the single-release setting.

**Theoretical Claims:**

The approach and the proof statements make sense. I have a few technical questions (below) that I could not easily answer myself. Though I did not check all details there seems to be no reason to suspect fundamental problems with correctness.

---

> ### Author Rebuttal · Authors · 2025-03-31
>
> We thank the reviewer for the detailed comments and valuable feedback.
>
> On using Misra Gries to track elements with high occurrency instead of blocklisting, we note that our algorithm’s goal is not to output elements with high occurrency but simply to filter them out at every timestep. While it is possible to implement a version of the Misra Gries algorithm that continually outputs items with high occurrency, our blocklisting method is simpler as it only involves sampling with a fixed probability.
>
> Thanks for your comment on the use of KSET vs other techniques like IBLT. We have considered using other more complex (and advanced set structures) — it is an interesting direction to pursue, in part because other methods like IBLT can reduce the space used by polylog factors. However, we chose to use KSET because for the DP guarantees, we need to open the black box of the algorithm and analyze its consistency properties (something that is not needed for non-private work). The largely consistent output for KSET in the presence of increased collisions makes it more amenable to analysis in DP settings. We stress that even our use of the simpler to analyze KSET method in the DP setting requires significant care especially when dealing with the failure cases.
>
> Q1: For the comment on the “output being a multiple of a power of 2”, first we observe that actually the output is the product of $2^i$ times a real value (a DP estimate from a count + gaussian noise) so it is not actually an integer power of two (even after rounding). This aspect is however not crucial for the accuracy proof (see below). The scaling up by a factor of $2^i$ is to account for the sampling rate of $2^{-i}$ in the $i$-th stream. In terms of how this relates to our accuracy guarantees (and in particular, our $(1+\eta)$-approximation guarantee), we note that our accuracy guarantee follows from the proof of Lemma A8 in appendix, which shows how our sampling rate is sufficient to give accuracy of the output of the substreams in terms of $(1+\eta)$ multiplicative error plus some additive error. This is because we only need the number of distinct elements in the downsampled stream to be within a multiplicative constant (specifically, 2) range for the concentration bounds to hold with the desired multiplicative approximation and with the allowed additive error. Notice as we discuss in the introduction of our paper, any low-space private algorithm for this problem must incur both multiplicative and additive error.
>
> Q2 & Q3: Regarding the hash functions (and whether they need to be fully random), our construction only requires $\lambda$-wise independent hash functions. Specifically, we use pairwise independent hash functions for the KSET subroutine and $\lambda$-wise independent hash functions for the CountDistinct subroutine. The space complexity for the latter is $O(\lambda \log T)$, which simplifies to $O(polylog(T))$, and thus remains efficient in our setting. We note that using universal hash functions might introduce dependencies as it only ensures low collision probability for any pair of elements, and this may affect utility. In particular, our utility analysis relies on a concentration bound (Lemma D.5) for $\lambda$-wise independent hash functions and thus would need to change if we use universal hash functions instead.

---

### Official Review · Reviewer_uSeR · 2025-03-14

**Overall Recommendation:** 3

**Summary:**

This paper introduces the first differentially private algorithm for counting distinct elements in the turnstile model to obtain *sublinear* space-complexity.
Their new approach is to introduce the notion of occurency (maximum number of times any element appears in a stream) and are able to achieve sqrt(W) additive error, where W is the occurency of the stream. For arbitrary streams they achieve additive error of T^{1/3}, where T is the length of the stream.

**Claims And Evidence:**

They provide proofs of their results.

**Essential References Not Discussed:**

No

**Experimental Designs Or Analyses:**

Not applicable.

**Methods And Evaluation Criteria:**

Not applicable.

**Other Comments Or Suggestions:**

I think it would be fitting to define \rho-zCDP in the main part of the paper instead of in the appendix, since this is the important property that is (at least directly) shown for the algorithm.

**Other Strengths And Weaknesses:**

The result of the paper is good, achieving simultaneously low memory, high accuracy and differential privacy in the streaming distinct elements problem. The authors do a good job explaining the previous work that motivated this result.
However, this paper has a lot going on and a lot of different subroutines only provided in the appendix, which can make it hard to understand how the overall algorithm works.
Also, one could argue, that counting distinct elements is not a core machine learning task, so there might be a question of scope. This is only a minor issue as ICML has traditionally published papers in this direction.

**Questions For Authors:**

No questions

**Relation To Broader Scientific Literature:**

Counting distinct elements in a data stream is an extremely well studied topic. This paper adds the requirement of differential privacy to the problem. This seems well motivated, especially since they are not the first to look at this.

**Theoretical Claims:**

Only a small part of the proofs are provided in main article and these refer to algorithms and lemmas in the appendix, so I cannot confidently claim to have checked correctness of these. I did not read proofs in appendix.

---

> ### Author Rebuttal · Authors · 2025-03-31
>
> We thank the reviewer for the comment about the subroutines and other technical details of the main algorithm that currently appear in the appendix. Based on this feedback, we plan to reduce the space dedicated to the technical overview (Section 2) in the final version, and instead use the space to incorporate additional core content from the appendix into the main body. We agree with the reviewer’s comment that this would improve readability of the main body. We will also move the definition of \rho-zCDP back to the main text for better clarity of this concept.
>
> As for the significance of the counting problem, we observe that counting serves as a fundamental primitive in the analysis of many private algorithms. Advancements in private counting – especially in the continual release setting – are crucial for the development of private machine learning. For instance, the state-of-the-art differentially private stochastic gradient descent (DP-SGD) underpinning privacy-preserving AI have consistently relied on advances in private sum queries over data streams (including the work on binary tree mechanisms and advanced correlated-noise counting schemes). We believe that foundational research on efficient DP continual release algorithms is necessary for advancing private ML.  We observe that the prior work “Counting Distinct Elements in the Turnstile Model with Differential Privacy under Continual Observation” Jain et al. published at NeurIPS 2023, also highlights the relevance and importance of this problem within the machine learning literature.

---

### Official Review · Reviewer_wC4M · 2025-03-18

**Overall Recommendation:** 4

**Summary:**

This work gives a new space-efficient algorithm for differentially private counting of distinct elements under the turnstile model of continual release. In this setting, the algorithm sees a length $T$ stream of insertions, deletions, or null updates of elements from a fixed domain. At each timestep of the stream, the algorithm releases the count of distinct elements remaining after the stream of operations. Their algorithms use $\log T$ dictionaries each of which hold at most $T^{1/3}$ items to approximate the counts, giving a $\tilde{O}(T^{1/3})$-space algorithm. This improves on the space complexity of prior work ($O(T)$ space), while still maintaining a competitive additive error bound of $O(T^{1/3})$.

**Claims And Evidence:**

All claims in the work are supported with proof.

**Essential References Not Discussed:**

The related work section missed a recent and very relevant paper, Private Counting of Distinct Elements in the Turnstile Model and Extensions by Henzinger, Sricharan, and Steiner 2024. This work studies the same problem and improves over the work of Jain, Kalemaj, Raskhodnikova, Sivakumar, and Smith 2023 in a few dimensions. It does not give a significant improvement in space complexity, however, so does not subsume this work.

**Experimental Designs Or Analyses:**

N/A

**Methods And Evaluation Criteria:**

N/A

**Other Comments Or Suggestions:**

On page 5 it’s stated that the parameter $k$ should be polylogarithmic in $T$, which was a bit confusing given that the space complexity ends up being $T^{1/3}$. I understand this to mean that for fixed occurrency not depending on $T$, $k$ should have only polylogarithmic dependence on $T$, so I don’t think it’s an unfair claim, but I think it somewhat misleads the reader as to what the role of the KSET dictionaries will be in the algorithm.

**Other Strengths And Weaknesses:**

Strengths: This paper shows how to use a series of dictionaries, each of which store elements subsampled from the stream at varying rates, to avoid maintaining the exact counts of all elements in the stream. This represents, to the best of my knowledge, a significant improvement in space over prior work, at least in the regime where the domain of stream elements is much larger than $T^{1/3}$.

Weaknesses: A comparison to the work mentioned above should be included in the work.

**Questions For Authors:**

Please see comments in strengths and weaknesses section.

**Relation To Broader Scientific Literature:**

With the exception of the paper mentioned below, and to the best of my knowledge, this paper adequately contextualizes its results with comparison to prior work.

**Theoretical Claims:**

I read all proofs contained in the body of the paper and found no issues. I did not, however, read the proofs in the appendices.

---

> ### Author Rebuttal · Authors · 2025-03-31
>
> We thank the reviewer for the thoughtful comments and valuable feedback.
>
> Regarding the comparison against Henzinger, Sricharan, and Steiner (2024), we do, in fact, compare our result to this work in Line 192 of the submitted version; however, we cited an earlier version of the paper, titled "Differentially Private Data Structures under Continual Observation" (2023). We will update the reference to the 2024 version in the final paper for better clarity in the references. We appreciate the reviewer bringing this to our attention.
>
> As for the statement on page 5 of “the space parameter k should be polylog(T)”, we apologize for the typo. More correctly we meant that the space should be sublinear in T, and in our paper it is indeed O(T^{1/3}), and we will make this correction in the final version of the paper.

---

> > ### Comment · Reviewer_wC4M · 2025-04-07
> >
> > My mistake about the citation! I've updated my score to reflect the change.

---

> > > ### Author Response · Authors · 2025-04-07
> > >
> > > Thank you for responding to our rebuttal and for updating your score!

---

### Decision · Program_Chairs · 2025-05-01

**Decision:**

Accept (poster)

**Comment:**

The paper presents a new space-efficient algorithm for differentially private counting of distinct elements under the turnstile model of continual release. In this setting, at each step of the execution, the algorithm observes the next update in the stream (an insertion or deletion) and releases an updated estimate of the current number of (remaining) distinct elements. The proposed algorithm improves the space complexity over prior work while maintaining a competitive additive error. The reviewers all agree that the results are strong and the techniques are interesting. This paper would be a valuable addition to ICML.